# Assessing the Relationship between the Gut Microbiota and Inflammatory Bowel Disease Therapeutics: A Systematic Review

**DOI:** 10.3390/pathogens12020262

**Published:** 2023-02-06

**Authors:** Cassandra Mah, Thisun Jayawardana, Gary Leong, Sabrina Koentgen, Daniel Lemberg, Susan J. Connor, Theodore Rokkas, Michael C. Grimm, Steven T. Leach, Georgina L. Hold

**Affiliations:** 1Microbiome Research Centre, School of Clinical Medicine, University of New South Wales, Sydney, NSW 2217, Australia; 2Discipline of Paediatrics, School of Clinical Medicine, University of New South Wales, Sydney, NSW 2052, Australia; 3Gastroenterology, Sydney Children’s Hospital Randwick, Randwick, NSW 2031, Australia; 4Ingham Institute for Applied Medical Research, South Western Sydney Clinical School, University of New South Wales, Sydney, NSW 2170, Australia; 5Department of Gastroenterology Hepatology, Liverpool Hospital, Sydney, NSW 2170, Australia; 6Gastroenterology Clinic, Henry Dunant Hospital, 11526 Athens, Greece

**Keywords:** inflammatory bowel disease, Crohn’s disease, ulcerative colitis, therapeutics, gut micro-biota

## Abstract

Current inflammatory bowel disease (IBD) treatments including non-biological, biological, and nutritional therapies aim to achieve remission and mucosal healing. Treatment efficacy, however, is highly variable, and there is growing evidence that the gut microbiota influences therapeutic efficacy. The aim of this study was to conduct a systematic review and meta-analysis to define changes in the gut microbiota following IBD treatment and to identify microbial predictors of treatment response. A systematic search using MEDLINE/Embase and PubMed was performed in July 2022. The review was conducted based on the Preferred Reporting Items for Systematic reviews and Meta-Analyses guidelines. Studies were included if they reported longitudinal microbiota analysis (>2 weeks) using next-generation sequencing or high-throughput sequencing of faecal/mucosal samples from IBD patients commencing treatment. Meta-analysis on alpha-diversity changes following infliximab treatment was conducted. Thirty-nine studies met the inclusion criteria, and four studies were included in the meta-analysis. An increase in alpha diversity was observed following treatment with 5-aminosalicylates, corticosteroids, and biological therapies in most studies. Characteristic signatures involving the enrichment of short-chain-fatty-acid-producing bacteria including *Faecalibacterium prausnitzii* and a reduction of pathogenic bacteria including various Proteobacteria were demonstrated following treatment with specific signatures identified based on treatment outcome. The meta-analysis demonstrated a statistically significant increase in bacterial richness following infliximab treatment (standardised mean difference −1.16 (−1.50, −0.83), *p* < 0.00001). Conclusion: Distinct microbial signatures are seen following treatment and are associated with treatment response. The interrogation of large longitudinal studies is needed to establish the link between the gut microbiota and IBD therapeutic outcomes.

## 1. Introduction

Inflammatory bowel diseases (IBDs), of which the main subtypes are ulcerative colitis (UC) and Crohn’s disease (CD), are chronic immune-mediated inflammatory conditions that require lifelong treatment [1]. The natural history of IBD is that of quiescent disease interspersed with flares in disease activity with significant socio-economic implications [2,3].

Current drug treatment strategies, which include 5-aminosalicylates (5-ASA), corticosteroids, and immunomodulators, aim to achieve remission and mucosal healing through reducing the inflammatory burden during active disease [4]. However, they do not specifically target the cause of the inflammation. Growing understanding of the complex multifactorial aetiopathogenesis of IBD has led to the development and use of more targeted therapies, specifically biological agents (anti-TNF-alpha antibodies, anti-cytokine monoclonal antibodies, and anti-integrin monoclonal antibodies) to treat moderate to severe disease [5,6,7,8].

Anti-TNF-alpha agents, including infliximab, are often first-line in the majority of biologic naïve IBD patients [1]; however, the efficacy and tolerability of these therapies are highly variable among patients. It is estimated that 33–50% of patients, especially in the elderly population, discontinue anti-TNF therapy, with a further 13–20% of them experiencing yearly loss of response [9,10,11]. Nutritional therapies including exclusive enteral nutrition (EEN), probiotics, prebiotics, and a specific carbohydrate diet (SCD) aim to induce and maintain disease remission, but mechanistic insight into the mode of action of nutritional therapy remains in its infancy.

IBD is thought to arise from defective immune regulation, producing an abnormal response to luminal antigens, which enter the intestinal wall via a leaky epithelium, thus triggering inflammation in susceptible individuals [12,13]. IBD risk variants often implicate pathways for microbial recognition or handling/clearance strategies. However, the rapid increase in disease burden in the last 100 years refutes the significance of genetic risk and elevates the importance of other environmental factors, which have changed more dramatically in line with disease prevalence.

The healthy human gut microbiota consists of four major bacteria phyla (Firmicutes, Bacteroidetes, Proteobacteria and Actinobacteria) with the gut microbiota composition shaped by genetics, ethnicity, environmental exposures, diet, and lifestyle factors [14,15,16]. Changes or imbalances in the gut microbiota, termed dysbiosis, have been consistently associated with IBD [17,18,19]. Studies have shown that dysbiosis imparts both community and functional changes, which drive a proinflammatory state [16].

The most-consistent finding in IBD, in terms of gut microbiota changes, is decreased bacterial diversity/richness, also referred to as alpha diversity. From a taxonomic perspective, IBD is associated with a reduction in members of the Firmicutes phyla, including beneficial bacterial species such as *Faecalibacterium prausnitzii*, and an increase in pathogenic proinflammatory species, including members of the Enterobacteriaceae including certain species of *Escherichia coli* and Bacteroidetes [20,21]. In CD specifically, reductions in *Roseburia* and *Faecalibacterium* genera and an increase in *Ruminococcus gnavus* have been consistently reported [22,23]. Specific features of UC-associated dysbiosis are not so consistent, although increases in sulphate-reducing bacteria have been observed [24,25]. Faecal microbiota transplant (FMT) as a successful UC treatment and FMT colitis mouse models highlight and support the central role of the gut microbiota in IBD pathogenesis [26,27].

Recent studies have established the ability of gut microbes to alter the efficacy of various therapeutics including cardiac drugs (digoxin), L-dopa treatment for Parkinson’s disease [28,29], and cancer treatments [30,31]. For example, a distinct baseline gut microbiota enriched with *F. prausnitzii* and other Firmicutes was associated with a beneficial clinical response to immune checkpoint inhibitor, ipilimumab, in the treatment of metastatic melanoma [31]. This key finding was corroborated by Gopalakrishnan and colleagues, who analysed the gut microbiome of melanoma patients undergoing PD-1 immunotherapy in which characteristic differences were observed in terms of the diversity and composition of the gut microbiome in responders versus non-responders [32]. These studies demonstrated the relationship between gut microbes and the efficacy of pharmacotherapies and highlighted the promising potential of the gut microbiota as a predictor or modulator of treatment response.

Similarly, in IBD, the potential role of the gut microbiota as a predictive biomarker for treatment response has been an area of intense research. It is currently unclear whether microbiota analysis is superior to existing parameters including C-reactive protein, erythrocyte sedimentation rate, the endoscopic disease activity index, and pathologic findings. Several recent longitudinal studies have analysed the impact of 5-ASA, corticosteroids, biological therapies, and nutritional therapies on the gut microbiota in IBD patients and their association with clinical outcomes [33,34,35,36,37,38,39,40,41,42]. Most studies have identified associations between unique microbial changes during treatments and favourable disease outcomes. However, whether these changes are sustained or treatment-specific remains unknown. Despite the growing number of studies on this topic, there has been no systematic review undertaken to date that has comprehensively summarised the impact of IBD therapeutics including nutritional therapies on the gut microbiota and, conversely, how the microbiota is related to treatment outcomes. The ability to predict treatment responses using personalised gut microbiota signatures would enable targeted, effective, and highly personalised treatment for optimal health outcomes for IBD patients. Therefore, the aims of this systematic review and meta-analysis were: (i) to define changes in the gut microbiota following IBD treatment and (ii) to identify microbial predictors of treatment response.

## 2. Materials and Methods

### 2.1. Search Strategy

This systematic review was registered at inception on PROSPERO, an international prospective register of systematic reviews (registration ID: CRD42021244046; 22 April 2021). The study was conducted based on the Preferred Reporting Items for Systematic reviews and Meta-Analyses (PRISMA) Guidelines [43]. A search of the online bibliographic databases PubMed (includes MEDLINE) and Embase was performed by two independent researchers (C.M. and G.L.) in April 2021 and updated in July 2022 (C.M. and T.J.). No restrictions were placed on the publication period. The following Medical Subject Heading terms were used, which included both the root terms and text words: inflammatory bowel disease, Crohn’s disease, ulcerative colitis, gastrointestinal microbiome, intestinal flora, dysbiosis, biological therapy, tumour necrosis factor antibody, infliximab, adalimumab, golimumab, certolizumab pegol, vedolizumab, ustekinumab, natalizumab, 5-aminosalicylic acid, mesalamine, sulfasalazine, corticosteroids, prednisone, budesonide, immunomodulation, immunomodulator, azathioprine, methotrexate, mercaptopurine, anti-bacterial agents, antibiotics, faecal microbiota transplantation, enteral nutrition, probiotic agent, prebiotic agent, specific carbohydrate diet. Synonyms and word variations were combined using the AND and OR functions. Manual searches of the reference list from potentially relevant review articles, published editorials, and retrieved original studies were performed to identify additional studies that may have been missed using the computer-assisted search strategy. The Covidence systematic review software was used during the screening process to identify the most-relevant papers.

### 2.2. Study Selection and Eligibility Criteria

Randomised controlled trials (RCTs), cohort studies, and observational studies reporting on microbiota changes in patient faecal samples and/or mucosal biopsies were evaluated. Eligible studies were included in this systematic review if they met all the inclusion criteria. Studies that reported duplicated results or where data could not be extracted were excluded.

Studies involving adults or children (>5 years old) previously diagnosed with IBD using clinical, endoscopic, and/or pathologic features and using a commonly accepted method to assess disease progression and outcomes were eligible for inclusion. Studies were included if subjects on IBD therapy were followed up for at least two weeks with microbial analysis of faecal or biopsy samples performed at baseline and the endpoint using next-generation or high-throughput sequencing. Microbial analysis reporting was required to provide information on the presence or abundance of microbial taxa.

Studies were excluded if they performed solely culture-based analysis, did not report on patterns of individual bacterial taxa differences, or if the microbial analysis did not include information on the baseline status prior to therapy. Additionally, case reports, studies of fewer than ten patients, studies focusing on children <5 years old, conference articles, review articles, animal studies, studies that did not provide information on antibiotic usage, or studies reported in languages other than English were excluded.

### 2.3. Outcome Assessment

The primary outcomes were (i) changes in the gut microbiome of IBD patients following commencement of treatment with biological therapies, 5-ASA, corticosteroids, antibiotics, or nutritional therapies and (ii) the identification of microbial predictors of treatment response.

### 2.4. Data Extraction and Analysis

The Newcastle and Ottawa scale (NOS) was used to assess the quality of the studies [44] Cohort studies were scored across 3 categories: selection (4 questions) and comparability (1 question) of study groups and ascertainment of the outcome of interest (3 questions) with all questions with a score of 1, except for the comparability of study groups, in which separate points were awarded for adjusting for confounders (maximum of 2 points). An aggregate score of 8 or greater was suggestive of a high-quality study. Different scales were used to assess cohort studies and RCTs.

### 2.5. Data Synthesis and Statistical Methods

Information from the included studies was extracted into a Microsoft Excel spreadsheet. Where available, the following clinical data were extracted for each trial: study type, country of origin, diagnosis, age, duration of disease, intervention and its dosage, duration of intervention, sample types, length of follow-up, method of microbiota composition analysis, and microbiota changes. Studies were eligible for the meta-analysis if more than two studies shared the same outcome and had reported usable data in a compatible metric. Revman5 (Review Manager (RevMan) (Computer program), Version 5.4.1, The Cochrane Collaboration, 2020) was used to conduct the meta-analysis. A meta-analysis of four studies involving 82 patients looking at the mean differences in alpha diversity measured by the Shannon index at baseline and after exposure to biologic treatment with anti-TNF-alpha (infliximab) was undertaken. Shannon index values were analysed using an inverse variance model with a 95% confidence interval. Values are reported as standardised mean differences (SMDs). *p*-values were two-tailed, and statistical significance was set at *p* < 0.05. The I2 test was used to quantify heterogeneity estimates, i.e., <30%, 30–59%, 60–75%, and >75% defined the thresholds for low, moderate, substantial, and considerable heterogeneity, respectively. When statistically significant heterogeneity was not present, pooled estimates (95% CI) were calculated using the fixed-effects model. If significant heterogeneity was present, pooled estimates (95% CI) were calculated using the random-effects model [45,46,47].

## 3. Results

### 3.1. Search Results

The search strategy identified 10,008 references that were imported for screening, and 1266 duplicates were removed. Three reviewers (C.M., G.L., and T.S.) independently performed an initial screen of the titles and abstracts based on the eligibility criteria, with 8479 articles excluded at this stage. Then, 263 studies were assessed for full-text eligibility. Of these, 205 were excluded, and of these, 95 were conference proceedings, 19 full texts that were not retrievable, 16 were review articles/study protocols, 20 reported on the microbiota without using high-throughput sequencing, 12 had insufficient microbial data, 11 lacked longitudinal data, 16 had insufficient data on intervention, 3 had the wrong intervention, 2 had the wrong outcomes, 1 included fewer than six patients, 1 did not specify treatment, 3 had the wrong patient population, and 1 was found to be a duplicate. Disagreements between reviewers were resolved by consensus with third-party experts (G.H. and S.L.). Twenty-eight studies on faecal microbiota transplantation were out of the scope of this systematic review and, therefore, excluded. Thirty-nine studies met the eligibility criteria and were included in the analysis (Figure 1).

### 3.2. Description of Included Studies

The characteristics of the studies involving biological therapies, non-biological therapies, and nutritional therapies stratified by disease subtypes (studies with CD patients, UC patients, or both) are summarised in Table 1. All thirty-nine studies were prospective in nature, with thirty-four cohort studies and five RCTs. All studies reported the impact of IBD therapeutics on alpha diversity and/or taxonomic changes by conducting microbial assessment at baseline prior to therapy initiation and at least two weeks post therapy exposure. Some studies also reported longer-term outcomes of more than a year [42,48,49]. Nineteen studies involved only CD patients; five studies involved only UC patients; fifteen studies included both UC and CD patients. Twenty-one studies focused on adult subjects, while eighteen studies evaluated paediatric cohorts. Twenty studies reported on biological therapies: anti-TNF-alpha therapies (infliximab, adalimumab, etanercept, golimumab; n = 18), anti-integrin antibody therapy (vedolizumab; n = 2), and anti-IL-12/23 monoclonal antibody therapy (ustekinumab; n = 1) (Table 1). One study assessed the impact of 5-ASAs, two corticosteroids, two the impact of antibiotics, and one study the impact of azathioprine (Table 1). There were fourteen studies reporting on nutritional therapies, specifically exclusive enteral nutrition (EEN; n = 8), specific carbohydrate diet (SCD; n = 2), prebiotics (n = 2), adsorptive granulomonocytapheresis (GMA; n = 1), and dietary intervention (n = 1).

Quality assessment using the NOS for observational cohort studies and RCTs was undertaken (Appendix A). Scores of 0–3, 4–6, and 7–9 were considered as low, moderate, and high quality, respectively. Of the cohort studies, fifteen scored the maximum 9 points, twelve scored 8 points, four scored 7 points, and three scored 6 points. All except three studies were deemed to be of high quality. Olbjorn et al. did not include information confirming that the use of the IBD treatment of interest was not present at the start of the study [50]. Valcheva et al. and Suskind et al. did not include information on the selection of the non-exposed cohort [51,52]. Similarly, no study control factors involving the IBD subjects were documented in the three moderate-quality studies. Of the RCTs, two studies attained 9 points, two studies scored 7, and one study scored 6 points, meaning four articles were deemed to be of high quality, while the remaining one was of moderate quality. The study of moderate quality did not have information on the representativeness of the controls and was not blinded, and the patients were recruited in a hospital setting [34].

**Table 1 pathogens-12-00262-t001:** Study characteristics of the studies that included CD patients only, UC patients only, or both, stratified by biological therapies, non-biological therapies, and nutritional therapies.

Disease Subtype	Study	Study Type	Country, Single-(SC) /Multi-Centre (MC)	Diagnosis (# Subjects)	Age Group	Age Group		Duration of Disease (in Years) (Mean SD)	Intervention (# of Subjects); Route, Dose	Length of Intervention (Sampling Points (Weeks))	Sample Type	Length of Follow-up (Weeks + Sampling Points)	Method of Microbial Composition Analysis
						Age (Years) Mean (SD)	Age (Years) Median/IQR						
	**Biological Therapies**												
CD CD	Wang et al. [39]	Prospective cohort study (PCS)	China, SC	CD (n = 11) HC (n = 16)	Paediatric	CD: ND (range 4–17)	ND	ND	Infliximab (IFX; n = 11); IV, 5mg/kg	0, 2, 6 + every 8 weeks	Faeces	ND	16S rRNA sequencing analysis (16S SA)
	Doherty et al. [41]	Randomised controlled trial (RCT)	U.S., MC	CD (n = 306)	Adult	CD: 39 (SEM 13)	ND	CD: 12 (SEM 8.8)	Ustekinumab (UST, n = 232); IV, ND (induction therapy (IT)) UST; SC, 270 mg or 90 mg SC (maintenance therapy (MT))	8 weeks (ND; IT) 8 weeks (8, 16; MT)	Faeces	0, 4, 6, 22	16S SA
	Ribaldone et al. [53]	PCS	Italy, SC	CD (n = 20)	Adult	ND	CD: 52.5 (range 26–69)	CD: 14.5, ND	Adalimumab (ADA, n = 20); ND	6 months (ND)	Faeces	0, 26	Metagenomic (MGS) and 16S SA
	Kowalska-Duplaga et al. [54]	PCS	Poland, ND	CD (n = 18) HC (n = 18)	Paediatric	ND	ND	ND	IFX (n = 18); 5 mg/kg, IT	0, 2, 6	Faeces	0, 14 and 6–8 weeks post drug)	16S SA
	Zhuang et al. [40]	PCS	China, SC	CD (n = 49)	Adult	CD: 25 (range 18–48)	ND	CD: 2 (median range 0.5–17)	IFX (n = 49); IV, ND	0, 2, 6, 14, 22, 30	Faeces	0, 6, 30	16S SA
	Salamon et al. [55]	PCS	Poland, SC	CD (n = 61) HC (n = 17)	Paediatric	CD on IFX: 13.09 (3.76) HC: 11.73 (2.88)		ND	IFX (n = 13); IV 5 mg/kg	0, 2, 6	Faeces	10 weeks (0, 10); 4 weeks after 3rd dose	qPCR analysis
	Wang et al. [38]	PCS	China, MC	CD (n = 18) HC (n = 12)	Paediatric	ND	CD: 12 (11,14) HC: 12 (11,13)	ND	IFX (n = 18); IV 5–10 mg/kg	0, 2, 6 + every 8 weeks	Faeces	>6 weeks (0, 6 or 30 weeks; after 3rd or 6th dose)	16S SA
	Ventin-Holmberg et al. [56]	PCS	Finland, SC	CD-R (n = 12) CD-NR (n = 18)	Paediatric	ND	CD-R: 13 CD-NR: 14 (ND)	Median CD-R: 1.4 CD-NR: 0.3	IFX (n = 30), IV, ND	0, 2, 6	Faeces	0, 2, 6	16S SA
IBD IBD	Kolho et al. [43]	PCS	Finland, SC	UC (n = 26) CD (n = 36) IBDU (n = 6) HC (n = 26)	Paediatric	ND	IBD: 15.5 (range 9.6–18.3) HC: 13.4 (range 9.7–18.3)	UC, CD, IBDU: 3.5 (median range 0–10.7)	IFX (n = 31); ND ADA (n = 1); ND	ND	Faeces	>6 weeks (0, 2, 6 + 1 during MT)	Phylogenetic microarray and qPCR analysis
	Ananthakrishnan et al. [42]	PCS	U.S., SC	UC (n = 43) CD (n = 42)	Adult	ND	ND	Remission (n = 31): 9.2 (7.2) No remission (n = 54):15.7 (12.7)	Vedolizumab (VEDO, n = 85); IV, 300mg infusion	0, 2, 6 + every 8 weeks	Faeces	0, 6, 14, 30, 54	MGS
	Zhou et al. [57]	PCS	China, SC	UC (n = 51) CD (n = 16) HC (n = 73)	Adult	UC: 41.75 (14.36) CD: 31.81 (12.67) HC: 30.07 (6.36)	ND	ND	IFX (n = 16); 5 mg/kg	0, 2, 6, 14, 22, 30	Faeces	0, 30	16S SA
	Aden et al. [58]	PCS	Germany, SC	Discovery Cohort (DC): UC (n = 4), CD (n = 8), HC (n = 19) Validation Cohort (VC): UC (n = 13), CD (n = 10), HC (n = 99)	Adult	DC- IBD: 45.4 (4.5) HC: 26.37 (0.5) VC; ND	ND	ND	IFX (n = 10); ND Etanercept (n = 12); ND VEDO (n = 13); ND	ND	Faeces	0, 2, 6, 30 0, 2, 6, 14	16S SA
	Olbjorn et al. [50]	PCS	Norway, MC	HC (n = 70) non-IBD (n = 50) CD (n = 80) UC (n = 27) IBDU (n = 3)	Paediatric	ND	ND	ND	Immunomodulators (n = 98); ND Anti-TNF therapy (n =6 4); ND	ND	Faeces	0, 78	16S SA
	Dovrolis et al. [59]	PCS	Greece, SC	UC (n = 6) CD (n = 14) HC (n = 9)	Adult	ND	ND	ND	IFX (n = 14); IV, 5 mg/kg	0, 2, 6 + every 8 weeks	Mucosal Biopsies	12–20 weeks (0, 12–20)	qPCR and 16S SA
	Ding et al. [48]	PCS	ND	CD (n = 76) UC (n = 10) HC (n = 13)	Adult	ND	CD: 38.49 14.61 (19–82) UC: 25.52 17.34 (19–70) HC: 30.5 2.94(27–35)	ND	IFX (n = 66); ND (IT) ADA (n = 10); ND (IT)	ND	Faeces	16 months (3 monthly)	16S SA
	Schierova et al. [60]	PCS	Czech Republic, SC	CD (n = 34, with 17 at endpoint (EP)) UC (n = 18, with 10 at EP) HC (n=37)	Adult	CD: 35.0 (26.5, 44.0) UC: 31.0 (26.0, 41.3) HC: 36.5 (28.8, 41.3)	ND	CD: 26 (22.5, 35) UC: 27 (21.5, 31)	Anti-TNF therapy (CD (n = 17), ND UC (n = 10), ND)	38	Faeces	0, 2, 8, 14, 20, 26, 32, 38)	16S SA
	Sanchis-Artero et al. [61]	Prospective observational study	Spain, MC	CD (n = 27) HC (n = 16)	Adult	CD: 41.4 (17.4) HC: 29.3 (7.2)	ND	ND	Anti-TNF therapy (n = 27), ND	26	Faeces	0, 13, 26	16S SA
	Ventin-Holmberg et al. [49]	PCS	Finland, SC	CD (n = 25, with 19 at EP) UC (n = 47 with 33 at EP)	Adult	ND	31 (24–45)	Median 2 (0–7)	IFX (n = 72, ND, ND)	52	Faeces	0, 2, 6, 12, 52	16S SA
	Effenberger et al. [62]	Prospective comparative study	Austria, SC	CD (n = 24) UC (n = 12)	Adult	CD: 33.9 (+/−12.9) UC: 44.1 (+/−14.5)	ND	ND	anti-TNF therapy	30	Faeces	0, 12, 30	16S SA
	Park et al. [63]	PCS	South Korea, MC	CD (n-10) UC (n = 9) HC (n = 19)	Adult	ND	IBD: 33 (23–52) HC: 31 (28–34)	ND	IFX (CD n = 7; UC n = 4) ADA (CD n = 3; UCn = 2) Golimumab (UC n = 3)	13	Faeces, saliva, serum, urine	0, 13	16S SA
**Non-Biological Therapies**													
CD	Pigneur et al. [34]	RCT	France, ND	CD (n = 19)	Paediatric	CD on corticosteroids: 13.7 (1.8) HC: ND	ND	ND	Corticosteroids (n = 6); ND, 1 mg/kg/ day (up to max of 60 mg)	4 daily IT + tapered over 3 months	Faeces	0, 8	16S SA
	Sprockett et al. [36]	RCT	Europe Canada, Israel, MC	CD (n = 67)	Paediatric	Metronidazole (MET): 13.5 (3.1) Metronidazole + Azithromycin (MET+AZ): 14.2 (3.1)	ND	MET: 0.7 (1) MET+AZ: 1.1 (1.1)	MET; n = 36); ND, 20 mg/kg twice daily (max 1000 mg/day) MET+AZ; n = 31 of which AZ, 7.5 mg/kg once a day (max 500 mg/day)	8 weeks (daily) + 4 weeks if lacked response	Faeces	0, 4, 8, 12	16S SA
UC	Ishikawa et al. [37]	PCS	Japan, SC	UC (n = 41)	Adult	UC on AFM: 44.7 (14.9)	ND	UC on FMT: 7.0 (8.0)	AFM monotherapy (n = 20);	2 weeks (daily)	Faeces	0, 4, 8	16S SA
	Schierova et al. [33]	RCT	Czech Republic, SC	UC (n = 16)	Adult	UC on 5-ASA: 40 (range 31–66)	ND	ND	Mesalazine (n = 8); enema, 4 g	6 weeks (daily for 2 weeks, every alternate day till week 6)	Faeces	0, 2, 4, 6, 12	16S SA
IBD	Hart et al. [35]	PCS	Canada, SC	UC (n = 10) CD (n = 20)	Paediatric	ND except CS ≤12 years (n = 7) CS ≥13 years (n = 7)	ND	ND	Methyprednisolone (n = 14); IV, 1 mg/kg/day (max 40 mg/day) Transition to oral corticosteroids 1 mg/kg/day PO x 2 weeks followed by progressive wean by 5 mg/wk	8 weeks (daily)	Faeces	0, 0.5, 1, 2, 4, 6, 8, 12	16S SA
	Effenberger et al. [62]	Prospective comparative study	Austria, SC	CD (n = 19) UC (n = 10)	Adult	CD: 49.2 (+/- 15.5) UC: 45.0 (+/−10.7)	ND	ND	Azathioprine	20	Faeces	0, 12, 30	16S SA
	**Nutritional therapies**												
CD	Kaakoush et al. [64]	PCS	Australia SC	CD (n = 5) HC (n = 5)	Paediatric	CD: 9.88 (ND) HC: ND	9.7	de novo presenting	EEN (OSMOLITE)	8–12	Faecal	0, 12 weeks post EEN	16S SA
	Quince et al. [65]	PCS	Scotland SC	CD (n = 23) HC (n = 21)	Paediatric	CD: ND (range 6.9–14.7) HC: ND (range 4.6–16.9)	NR	NR	EEN (Modulen)	8	Faecal	−1, 16, 32, 54	16S SA
	Tang et al. [66]	PCS	China	CD (n = 31) HC (n = 12)	Paediatric	HC n = 12; 10.45 ± 2.39; CD RE n = 17; 12.02 ± 1.88; CD NRE n = 8; 9.84 ± 4.51	NR	de novo presenting	EEN (oral)	8	Faecal	8 weeks	16S SA
	Dunn et al. [67]	PCS	Canada	CD (n = 10) HC (n = 5)	Paediatric	CD: 12.27	CD: 12	mainly de novo presenting	EEN; (nasogastric/gastric)	12	Faecal	12 weeks	16S SA
	Costa-Santos et al. [68]	PCS	Portugal	CD patients receive pre-operative EEN = 10; CD patients receive immediate surgery = 5	Adult	CD: 45.4 ± 19.1	NR	9 years (0–33 years)	Pre-operative EEN (oral)	15–70 days	Faecal	24 weeks	16S SA
	Diederen et al. [69]	PCS	Netherlands	CD (n = 27) HC (n = 18)	Paediatric	NR	CD: 14 (12–15) HC: 13 (11–16)	de novo presenting	EEN + concomitant thiopurines	6	Faecal	6 weeks	16S SA
	Tang et al. [66]	Prospective comparative study	China, SC	CD (n = 31) HC (n = 12)	Paediatric	CD:11.3 (3.5) HC:10.5 (2.4)	ND	ND	EEN (n = 31); oral, ND	0, 2, 8	Faeces	0, 2, 6, 8	16S SA
	Jiang et al. [70]	Prospective study	China, SC	CD (n = 7)	Adult	27.9 (12.4)	ND	ND	Enteral nutritional powder (n = 7); oral, 250 mL	0, 8	Faeces	0, 8	16S SA
	Suskind et al. [71]	RCT	U.S.	CD (n = 10)	Paediatric	14.3 ±2.9 (7–18)	NR	NR	Specific carbohydrate diet (SCD)	12	Faecal	NR	16S SA
UC	Wilson et al. [72]	Prospective study	U.K., SC	UC (n = 17)	Adult	35 (10)	ND	ND	GOS supplement (n = 17); oral, 2.8 g	0, 6	Faeces	0, 6	16S SA
	Chen et al. [73]	Prospective comparative study	China, SC	UC (n = 14) HC (n = 14)	Adult	UC and HC: 42.4 (15.0)	ND	ND	GMA (n = 28); intravenous, ND	5 “sessions”; (baseline and endpoint) (n = 5) 10 “sessions”; (baseline and endpoint) (n = 8)	Faeces	baseline and endpoint	16S SA
	Valcheva et al. [51]	PCS	Canada	UC (n = 25) Inulin 7.5g/d n = 12. Inulin 15g/d n = 13	Adult	Inulin 7.5g/d = 36 Inulin 15g/d = 39	NR	NR	Prebiotic (inulin)	9	Faecal	NR	16S SA
IBD	Suskind et al. [52]	Observational study	U.S., MC	CD = 9, UC = 3	Paediatric	12.8 ± 2.2.(10–17)	NR	1.3 ± 1.6 years (0–5)	Specific carbohydrate diet (SCD)	12	Faecal	NR	16S SA
	Olendzki et al. [74]	Prospective comparative study	U.S., SC	CD (n = 14) UC (n = 7)	Adult	40.5 (12.8)	ND	ND	Dietary intervention (n = 25); oral	8	Faeces	0, 8	MGS

Abbreviations: UC, ulcerative colitis; CD, Crohn’s disease; IBDU, inflammatory bowel disease unclassified; HC, healthy controls; qPCR, quantitative polymerase chain reaction; 16S SA, 16S rRNA sequence analysis; IV, intravenous, ulcerative colitis; CD, Crohn’s disease; IBDU, inflammatory bowel disease unclassified; HC, healthy controls; qPCR, quantitative polymerase chain reaction; 16S rRNA, 16S ribosomal RNA; MGS, metagenomic sequencing; IV, intravenous; EEN, exclusive enteral nutrition; GOS, galactooligosaccharide; GMA, granulomonocytapheresis; ND, not disclosed; NR, not reported, SC, single-centre; MC, multi-centre; IT, induction therapy; MT, maintenance therapy; RCT, randomised controlled study; PCS, prospective cohort study, IFX, Infliximab; ADA, Adalimumab, VEDO, Vedolizumab, UST, Ustekinumab; EP, endpoint; AFM, Amoxicilin; oral, 1500 mg/day; Fosfomycin; oral, 3000 mg/day, Metronidazole; oral 750 mg/day).

### 3.3. Impact of IBD Treatment on the Gut Microbiota

#### 3.3.1. Biologics

Of the twenty studies on biological therapies, eight studies included CD patients only and twelve studies enrolled both CD and UC patients. Four studies were conducted in China; three studies were from Finland; two studies were from Poland and the United States, respectively. Single studies originated from the United Kingdom, Greece, Italy, Czech Republic, Austria, Germany, South Korea, Spain, and Norway. Eighteen studies evaluated the impact of anti-TNF-alpha agents including infliximab, adalimumab, golimumab, and etanercept; two studies investigated the role of anti-integrin antibody (vedolizumab) and interleukin antibody (ustekinumab), respectively (Table 1).

A meta-analysis was carried out on the prospective cohort studies that examined the effects of infliximab on the gut microbiota in CD patients, measured by the Shannon index [38,39,40,54]. Eighty-two subjects were included in the meta-analysis that demonstrated a statistically significant increase in alpha diversity following commencement of infliximab treatment (SMD −1.16 (−1.50, −0.83), *p* < 0.00001; Figure 2). The degree of variance resulting from between-study heterogeneity, reported as I^2^, was 65%.

##### Microbial Diversity

Microbial species diversity in terms of alpha diversity was reported in 16 studies (Table 2). Measures of alpha diversity among studies varied with the use of the following: Shannon, Simpson, inverse Simpson, observed operational taxonomic units (OTUs)/species, observed number of phylotypes, Chao estimated richness, phylodiversity, Good’s coverage, Faith’s phylogenetic diversity, Pielou’s measure of species evenness indexes, ACE index, and Jackknife diversity. Of the 14 alpha diversity measures, the Shannon index was used in 10 studies, the inverse Simpson index in 5 studies, Chao1 index in 5 studies, and observed species in 4 studies. Raw alpha diversity data were only reported for eleven studies, and data from the remaining studies were obtained through manual extrapolation of published graphs.

Sixteen studies reported the impacts of biological therapies on alpha diversity with fifteen studies focusing on anti-TNF-alpha agents and one on interleukin inhibitor, Ustekinumab. Eight studies included only CD patients; one study included only UC patients, while seven studies included both UC and CD patients. Two studies demonstrated a statistically significant increase in alpha diversity [40,58]. Six studies involving CD patients and one study assessing an IBD cohort observed non-statistically significant changes in all measures of alpha diversity measured [43,59]. Dovrolis et al., who analysed both CD and UC patients, respectively, found reduced alpha diversity (Pielou index) in CD patients (regardless of treatment outcome) and UC responders on infliximab [59]. Similarly, Kolho and colleagues reported the same trend in non-responders on infliximab/adalimumab when measured with the inverse Simpson index [43]; however, they did not stratify their findings based on IBD subtypes. The statistical significance of alpha diversity changes compared to baseline were not reported in four studies [38,43,54,58]. One study evaluating the impacts of ustekinumab reported an increase in alpha diversity (measured by the inverse Simpson index) in both responder and non-responder groups [41]. A statistically significant increase compared to the baseline was only achieved at Week 22 in the responder group (Table 2).

In summary, increased alpha diversity following treatment was associated with response to infliximab, adalimumab, and ustekinumab [41,43,57]. Based on the included studies, the study period varied between 6 and 30 weeks. Despite the variability in the study periods, both the short- and longer-term data support the notion that an increase in alpha diversity is characteristic of patients commencing biologics.

**Table 2 pathogens-12-00262-t002:** Alpha diversity changes following biological therapies, non-biological therapies, and nutritional therapies in studies with CD patients only, UC patients only, or both.

Disease Subtype	Reference/No. of Subjects	Alpha Diversity Changes	Pre- Treatment (Week)	Post-Treatment (Week)			
**Biological therapies**							
**CD**	Wang et al. [38] CD (n = 4) HC (n = 16)	Week	0 *	14 *	22 *	30 *	38 *
		IFX Shannon#○	2.26	3.04	3.63	3.27	2.99
		IFX Observed Species#◊	148	219	321.5	281	202
		IFX Simpson#○	0.824	0.912	0.940	0.918	0.907
		IFX Inverse Simpson#○	5.68	11.4	18.3	12.1	10.8
		HC Shannon#○	3.49				
		HC Observed Species#◊	393				
		HC Simpson#○	0.898				
		HC Inverse Simpson#○	10.00				
	Kowalska-Duplaga et al. [54] CD (n = 18) HC (n = 18)	Week	0	14			
		IFX Shannon#○	3.762	4.326			
		IFX Observed Species#◊	46.5	64			
		IFX Faith PD#□	6.5723	7.65354			
		IFX Pielou#○	0.70198	0.6995			
		HC Shannon#○	4.664				
		HC Observed Species#◊	64				
		HC Faith PD#□	8.1631				
		HC Pielou#○	0.7739				
	Zhuang et al. [40] CD (n = 49)	Week	0	6	30		
		IFX Shannon#○	**2.43**	**2.61**	**2.87**		
		IFX Observed Species#◊	**113**	**145**	**166.5**		
		IFX Chao1#◊	**148.1**	**183.1**	**209.7**		
		IFX Good’s Coverage#◊	0.999	0.999	0.999		
	Wang et al. [39] CD (n = 11) HC (n = 12)	Week	0	T1^			
		IFX Shannon#○	3.46	3.62			
		IFX Chao1#◊	489.97	506.2			
		IFX Simpson#○	0.891	0.924			
		IFX Inverse Simpson#○	9.19	13.17			
		HC Shannon○	3.7	-			
		HC Chao1◊	475	-			
		HC Simpson○	0.94	-			
		HC Inverse Simpson○	14.5	-			
	Zhou et al. [57] CD R (n = 9) CD Relapse (n = 7) HC (n = 11)	Week	0	30			
		IFX R Shannon○	5.5	6.25			
		IFX R PD□	15	19			
		IFX Relapse Shannon○	5.5	5.6			
		IFX Relapse PD□	15	16			
		HC Shannon○	5.6	-			
		HC PD□	16	-			
	Doherty et al. [41] CD R (n = 18) CD NR (n = 30)	Week	0	4	6	22	
		UST R Inverse Simpson#○	6.65	9.44	8.42	**10.7**	
		UST NR Inverse Simpson○	5.25	5.0	5.5	6.5	
	Dovrolis et al. [59] CD R (n = 5) CD NR (n = 5)	Week	0	12–20			
		CD IFX R Pielou○	0.74	0.73			
		CD IFX NR Pielou○	0.74	0.725			
	Sanchis-Artero et al. [61] CD R (n = 13) CD NR (n = 14) HC (n = 16)	Week	0	24			
		CD R anti-TNF Shannon○	5.65	6.15			
		CD NR anti-TNF Shannon○	5.65	5.2			
		CD R anti-TNF Chao1◊	92.5	112.5			
		CD NR anti-TNF Chao1◊	92.5	68.8			
		HC Shannon○	6.33	ND			
		HC Chao1◊	116.3	ND			
**UC**	Dovrolis et al. [59] UC R (n = 2) UC NR (n = 2)	Week	0	12–20			
		UC IFX R Pielou○	0.74	0.69			
		UC IFX NR Pielou○	0.74	0.78			
**IBD**	Kolho et al. [43] IBD R (n = 6) IBD NR (n = 5)	Week	0	2	6		
		IFX/ADA Responder Inverse Simpson○	173	188	175		
		IFX/ADA Non-Responder Inverse Simpson○	133	113	128		
	Aden et al. [58] IBD (n = 12) HC (n = 19)	Week	0	2	6	30	
		Etanercept Shannon○	2.4	2.5	2.7	2.7	
		Etanercept Observed Species◊	**64**	**64**	**61**	**70**	
		Etanercept Chao1◊	**75**	**76**	**80**	**86**	
		Etanercept PD□	**6**	**6.5**	**6**	**7.5**	
		HC Shannon○	3.2				
		HC Observed Species◊	72				
		HC Chao1◊	86				
		HC PD□	7.5				
	Schierova et al. [60] CD (n = 26) UC (n = 18) HC (n = 37)	Week	0	20	38		
		CD Anti-TNF Shannon○	5.65	5.7	5.9		
		UC Anti-TNF Shannon○	5.2	5.85	5.85		
		HC Shannon○	6.15	ND	ND		
	Park et al. [63] IBD (n = 19) HC (n = 19)	Week	0	12			
		IBD Anti-TNF Shannon○	3.28	3.35			
		IBD Anti-TNF Simpson○	0.068	0.068			
		IBD Anti-TNF Chao1◊	290	315			
		IBD Anti-TNF ACE◊	305	325			
		IBD Anti-TNF Jackknife	295	330			
		IBD Anti-TNF NPShannon○	3.29	3.36			
		HC Shannon○	3.73	ND			
		HC Simpson○	0.05	ND			
		HC Chao1◊	465	ND			
		HC ACE◊	485	ND			
		HC Jackknife	503	ND			
		HC NPShannon○	3.73	ND			
	Ventin-Holmberg et al. [49] CD R (n = 13) CD PR (n = 4) CD NR (n = 6) UC R (n = 31) UC PR (n = 8) UC NR (n = 8)	Week	0	2	6	12	52
		CD R IFX Simpson○	9	9	12	8.75	10.5
		CD NR IFX Simpson○	7	8.5	8.3	9	ND
		CD PR IFX Simpson○	5.5	5.5	10.3	8	8
		CD R IFX OTU◊	97.5	105	107.5	107.5	110
		CD NR IFX OTU◊	90	95	97.5	95	ND
		CD PR IFX OTU◊	55	77.5	107.5	95	107.5
		UC R IFX Simpson○	11.25	11	10.5	10.5	11
		UC NR IFX Simpson○	7	10.75	8	10.25	4.75
		UC PR IFX Simpson○	9.75	8.5	10	10	10.5
		UC R IFX OTU◊	105	115	107.5	105	105
		UC NR IFX OTU◊	90	105	97.5	105	102.5
		UC PR IFX OTU◊	107.5	112.5	110	105	105
	Ventin-Holmberg et al. [56] IBD R (n = 10) IBD NR (n-16)	Week	0	2	6		
		IBD R IFX Simpson○	8.5	7	6.5		
		IBD NR IFX Inverse Simpson○	6	6.5	7		
		IBD R IFX OTU◊	75	72.5	72.5		
		IBD NR IFX OTU◊	65	66	70		
	Effenberger et al. [62] IBD R (n = 18) IBD NR (n = 18)	Week	0	98 days			
		R Anti-TNF Shannon○	0.4	0.2			
		NR Anti-TNF Shannon○	−0.2	0.175			
**Non-biological therapies**							
CD	Sprockett et al. [36] ^1 MET (n = 36) MET+AZ (n = 31) MET/MET+AZ (n = 11)	Week	0	4	8	12	
		CD MET Faith’s PD□	18.5	11.5	12	16	
		CD MET + AZ Faith’s PD□	**17.5**	**11**	**10**	**13**	
		CD MET/MET+AZ Faith’s PD□	**18.5**	**15.5**	**6.5**	**9.5**	
UC	Schierova et al. [33] UC R (n = 4) UC NR (n = 4)	Week	0	2			
		UC R 5-ASA Shannon○	6.6	6.9			
		UC NR 5-ASA Shannon○	6.65	6.85			
IBD	Hart et al. [35] ^2 UC (n = 10) CD (n = 4)	Week	0	12			
		Corticosteroids Shannon○	**2.85**	**3.6**			
	Effenberger et al. [62] IBD R (n = 15) IBD NR (n = 14)	Week	0	14			
		R AZA Shannon○	−0.2	−0.2			
		NR AZA Shannon○	−0.05	−0.85			
**Nutritional therapies**							
CD	Kaakoush et al. [64] CD (n = 5) HC (n = 5)	Week	0	8	26		
		EEN CD Shannon#○	2.25 ± 0.24	ND	ND		
		HC Shannon#○	2.75 ± 0.14	ND	ND		
	Quince et al. [65] CD (n = 23) HC (n = 21)	Week	0	2	4	8	Free diet
		EEN CD Shannon#○	2.88 ± 0.597	2.82 ± 0.492	2.49 ± 0.731	2.84 ± 0.513	3.04 ± 0.686
		HC Shannon○	ND				
	Tang et al. [66] HC (n = 12) CD Remission (n = 17) CD Non-remission (n = 8)	Week	0	8			
		EEN CD-R Shannon#○	1.84 ± 0.16	2.45 ± 0.1			
		EEN CD-NR Shannon#○	1.33 ± 0.23	1.76 ± 0.23			
		HC Shannon#○	3.1 ± 0.2	ND			
	Dunn et al. [67] CD (n = 10 at baseline, n = 9 at 12 weeks) HC (n = 5)	Week	0	12			
		EEN CD Sustained Remission EEN Chao	~1300	~1250			
		EEN CD Non-Sustained Remission Chao◊	~1000	~ 700			
		HC Chao◊	~2000	ND			
	Costa-Santos et al. [68] CD (n = 15)	Week	0	8			
		CD Faith’s PD#□	8 ± 2.3	5.2 ± 1.5			
	Diederen et al. [69] CD (n = 27) HC (n = 18)	Week	0	1–5	6	9	
		EEN CD Inv Simpsons#○	19.40	11.57	12.85	16.87	
		HC Inv Simpsons#○	20.37				
	Suskind et al. [52] CD (n = 8)	Week	0	12			
		SCD CD Shannon#○	2.62(2.4–3.1)	2.7(2.3–3.1)			
	Suskind et al. [71] CD (n = 5)	Week	0	2	12		
		SCD CD Inverse Simpson#○	103 (89–132)	115 (95–130)	112 (107–147)		
	Tang et al. [66] CD (n = 31) HC (n = 12)	Week	0	2	8		
		HC Shannon#○	3.1	ND	ND		
		EEN NR CD Shannon#○	1.33	ND	1.76		
		EEN R CD Shannon#○	1.84	ND	2.45		
		Observed Species#◊	172	ND	ND		
	Jiang et al. [70] CD (n = 7)	Week	0	8			
		Shannon#○	ND				
		Observed Species#◊	292	390			
UC	Wilson et al. [72] UC (n = 17)	Week	0	6			
		Shannon#○	2.5	2.2			
		Chao1#◊	50.4	35.6			
		Observed Species#◊	178	ND			
	Chen et al. [73] UC (n = 14) HC (n = 14)	*“Sessions”*	0	5	10		
		UC Shannon#○	ND				
		HC Shannon#○	ND				
		UC Observed Species#◊	523	751	521		
IBD	Olendzki et al. [74] CD (n = 14) UC (n = 7)	Week	0	6			
		Shannon#○	CD = 2.1 UC = 2.3				

# Reflects accurate alpha diversity values; * [38]: Time points are estimated based on IFX dosing schedule: Weeks 0, 2, and 6 followed by maintenance every 8 weeks. Patients give a baseline sample and one at various time points after IFX infusion; ^ [39]: T1 represents sampling post-treatment (3rd or 6th IFX infusion). IFX is given at Weeks 0, 2, and 6 followed by maintenance every 8 weeks; **Bold**: statistically significant compared to baseline (*p*-value < 0.05); # Reflects accurate alpha diversity values; ^1: In Sprockett et al., treatment ended at Week 8, and microbial analysis was investigated till Week 12 (4 weeks post-treatment); ^2 [35]: Treatment ended at Week 8, and microbial analysis was evaluated till Week 12 (day 84); ◊ Richness alpha diversity indices; observed species, Chao1, Good’s coverage, abundance-based coverage estimator (ACE), operational taxonomical units (OTU); ○ Abundance alpha diversity indices; Simpson, inverse (Inv) Simpson, Shannon, Pielou; □ Phylogenetic distance alpha diversity index; Faith’s phylogenetic distance (PD); Abbreviations: UC, ulcerative colitis; CD, Crohn’s disease; HC, healthy controls; EEN, exclusive enteral nutrition; SCD, specific carbohydrate diet; NR, non-responders; R, responders; PR, partial-responders; 5-ASA, 5-aminosalicyclic acid; MET, Metronidazole; AZ, Azithromycin; IFX, infliximab; ADA, adalimumab; UST, ustekinumab; anti-TNF, anti-tumour necrosis factor.

##### Microbial Composition Changes

Overall, a consistent reduction in the relative abundance of Proteobacteria including *Escherichia coli* and *Enterococcus* species and enrichment of more beneficial genera including short-chain fatty acid (SCFA) producers such as *Blautia* and *Faecalibacterium* species were observed following biological therapy in the CD, UC, and IBD cohorts [38,39,41,42,43,53,58,59]. Most studies investigated microbial changes in faecal samples of CD patients treated with infliximab [38,39,40,53,54,57,59]. Wang and colleagues conducted two similarly designed studies analysing the effects of infliximab in paediatric CD patients. In the earlier study (2018), the authors reported a sustained increase of SCFA-producing genera (*Blautia*, *Faecalibacterium*, *Odoribacter*, and *Sutterella*) and decreased abundance of *Enterobacteriaceae*, *Enterococcaceae*, *Planococcaceae*, and *Streptococcaceae* post-treatment [38]. Similar increases in the abundance of *Blautia* were also reported in their later study; however, Clostridium Cluster IV, *Collinsella*, *Eubacterium*, and *Ruminococcus* species were also increased. The study also observed a decreased abundance of *Abiotrophia* and *Lactococcus* species [39]. The findings by Zhuang and colleagues, in an adult CD cohort, corroborated the results reported by Wang et al., with a sustained enrichment of SCFA-producing taxa (*Lachnospira*, *Roseburia* and *Blautia*), as well as a reduction in pathogenic bacteria such as *Enterobacteriaceae* including *Escherichia-Shigella* and Fusobacterium [40]. Interestingly, in the only study evaluating mucosal biopsies of adult CD patients following infliximab treatment, Dovrolis and colleagues observed an increased abundance of Proteobacteria and *Rubrobacter*, which is in disagreement with other studies, thus highlighting the different microbial composition of stool versus mucosal samples [59]. In a small study involving both CD and UC patients, the restoration of normal levels of 14 phylotypes including several SCFA-producing bacteria following etanercept treatment was noted [58].

Doherty et al. evaluated microbiota changes following the use of ustekinumab on CD patients [41]. The study reported an increase in Bacteroidetes species, as well as *F. prausnitzii*, *Blautia*, *Ruminococcaceae*, and *Roseburia* species, which agrees with the anti-TNF-a studies. Following vedolizumab treatment, a reduction in *Bifidobacterium longum*, *Eggerthella*, *Ruminococcus gnavus*, *Roseburia inulinivorans*, and *Veillonella parvula* species’ abundance were observed between baseline and Week 14 post-treatment in CD patients. These changes were sustained over the 54-week study period [42].

##### Microbial Predictors of Treatment Outcomes

The association between microbiota changes following biological therapies and treatment outcomes was investigated in seven studies (Table 3). Five studies exploring associations between faecal microbiota profiles and infliximab therapeutic response identified microbial features that distinguished treatment responders from non-responders. Paediatric CD patients with sustained response had a higher abundance of SCFA-producing taxa, *Blautia*, *Faecalibacterium*, *Lachnospira*, and *Roseburia* compared to non-sustained responders [38]. In a more recent study by the same authors, an increased abundance of *Actinomyces*, *Atopobium*, and *Parabacteroides* genera were observed in patients with sustained treatment response [39]. Interestingly, the authors also demonstrated an association between baseline microbial composition and sustained treatment response. A higher abundance of *Methylobacterium*, *Sphingomonas*, *Staphylococcus*, and *Streptococcus* genera at baseline was associated with sustained response, with an increased abundance of Clostridium XI and Clostridium XVIII members, *Eggerthella*, *Lachnospiraceae incertae sedis*, *Parabacteroides*, and *Peptococcus* genera at baseline associated with a loss of response [39]. Increased proportions of *Lachnospiraceae* and *Blautia* genera at Week 6 following infliximab treatment were associated with clinical and endoscopic response in an adult CD cohort, whilst CD patients who had a relatively higher abundance of Clostridiales at baseline responded better to infliximab compared to those with lower abundance [49,57]. Clostridiales levels were also restored to almost healthy levels in those achieving clinical remission [61].

In contrast, in the study that analysed mucosal biopsies, the authors identified enterotypes distinct for responders and non-responders, which was inconsistent with the findings of faecal-based studies [59]. A high abundance of *Hungatella*, *Ruminococcus gnavus*, and *Parvimonas* genera at baseline were associated with clinical and endoscopic response, while a higher abundance of *Blautia*, *Faecalibacterium*, *Roseburia*, and *Negativibacillus* genera in CD patients at baseline was associated with non-response determined by clinical and endoscopic parameters. Increased prevalence of *Chloroflexi*, *Ruminococcus*, *Eubacterium hallii*, *Eubacterium eligens*, *Escherichia*, *Shigella*, and *Butyricicoccus* genera and decreased abundance of Fusobacteria were noted in CD patients who responded to treatment. In UC, responders had increased Bacteroidetes populations, *Veillonella*, *Tyzzerella*, *Ruminococcus torques*, *Parabacteroides*, *Erisipela*, *Clostridium*, and *Bilophila* genera, and loss of Spirochaetes and Plantomycetes. Non-responders had an increased abundance of Actinobacteria, *Porphyromonas*, *Granulicatella*, and *Corynebacterium* genera and a reduction in *Anaerostipes* abundance after 3 months of infliximab [59].

Higher abundance of *Bifidobacterium*, *Clostridium colinum*, *Eubacterium rectale*, an uncultured *Clostridiales*, and *Vibrio* species, as well as decreased abundance of *Streptococcus mitis* at baseline were associated with response to infliximab or adalimumab treatment compared to non-responders [43]. Similar findings were seen in adult CD patients on ustekinumab, with *Bacteroides*, *Faecalibacterium*, *Blautia*, *Ruminococcaceae*, and *Roseburia* species significantly more abundant in subjects in remission compared to those with active disease at 6 weeks post-treatment initiation [41]. In a study involving infliximab refractory CD patients switched onto anti-integrin, vedolizumab, a higher abundance of *Roseburia inulinivorans* and *Burkholderiales* genera at baseline was associated with remission at Week 14 compared to non-remitters [42]. Further analysis at 1 year post-treatment revealed that these specific microbial changes observed in patients that achieved remission at Week 14 were sustained and the treatment effect was durable.

#### 3.3.2. Non-Biological Therapies

##### Microbial Diversity

Seven studies evaluated the impact of non-biologic therapeutics on alpha diversity, and adequate data were available for extraction from four studies (Table 2). Schierova et al. reported that 5-ASA use in UC patients was associated with an increase in alpha diversity after 14 days of treatment [33]. The extent of alpha diversity increase was greater in treatment responders than non-responders; however, the results did not reach statistical significance. Hart et al. demonstrated that corticosteroid treatment was associated with a statistically significant increase in alpha diversity (Shannon index) in a cohort of CD and UC patients [35]. The increase in alpha diversity measured from baseline continued throughout 8 weeks of therapy and persisted at Week 12. Sprockett and colleagues evaluated the therapeutic impact of three antibiotic regimens: (1) Metronidazole (MET), (2) MET, and Azithromycin (AZ) or (3) initial MET and the combination of MET+AZ over a 12-week period in a CD cohort [41]. Alpha diversity, measured by Faith’s PD index, decreased over 8 weeks of treatment, rebounding slightly at Week 12 in all groups. No significant changes in the distance between baseline and Weeks 12 or 30 in either patients in remission or without remission were noted following azathioprine treatment [62].

##### Microbial Composition

There were six studies on non-biological therapies. Pigneur et al. observed an enrichment of SCFA-producing genera including *Roseburia intestinalis*, *Eubacterium species*, *Ruminococcus*, and *Bifidobacterium bifidum*, along with reduced numbers of *Blautia* species at Week 8 following corticosteroid treatment in a small paediatric CD study [34]. A similar increase in uncharacterised *Ruminococcaceae* was observed in paediatric UC and CD patients, which also reported an increased abundance of *Blautia* and *Sellimonas* species and a decreased abundance of *Granulicatella*, *Haemophilus*, and *Streptococcus* genera at Week 12 compared to baseline [35].

Antibiotic treatment is known to induce an ecosystem-wide disturbance and decrease microbial diversity. A Japanese study evaluated the impact of combination antibiotic regimen (amoxicillin, fosfomycin, and metronidazole) in nineteen adult patients with active UC [37]. The treatment resulted in an increased abundance of Proteobacteria, particularly *Enterobacteriaceae*, and a near complete depletion of Bacteroidetes species. Some recovery in the proportion of Bacteroidetes species was observed at both 4 and 8 weeks post-treatment [37]. A large multinational longitudinal study involving CD paediatric patients investigated the microbial impacts of two specific antibiotic regimens, MET and MET+AZ [36]. The finding of increased abundances of potentially pathogenic *Enterococcus* in the MET group concurred with the Japanese study’s findings, which may indicate the presence of antibiotic resistant microbes and their enrichment predisposing patients to invasive infections. Interestingly, the opposite effect was observed in the MET+ AZ treatment, which might be reflective of the effects of AZ against adherent and invasive *E. coli* strains [36]. The 5-ASA study failed to observe specific microbial changes over the 12-week study period [33]. CD patients not benefiting from azathioprine treatment had increased abundance of *Lactobacillus* and *Klebsiella* genera [62]. In contrast, AZA responders had a significant decrease in Proteobacteria and an associated increase in Bacteroidetes (Table 3).

#### 3.3.3. Nutritional Therapies

Of the fourteen studies on nutritional therapies, five studies were conducted in North America, four studies in China, three studies in Europe, and one study in Australia and Canada, respectively. Nine studies involved solely CD patients; three studies enrolled UC patients; two involved both UC and CD patients. Eight of the studies evaluated the impact of EEN (6 paediatric and 2 adult cohorts); two paediatric studies assessed the impact of SCD; two adult studies assessed the impact of the prebiotic inulin and galactooligosaccharide (GOS) supplement, respectively; one study assessed adsorptive (GMA); another studied dietary intervention on adult cohorts (Table 1).

##### Microbial Diversity

The results of eight studies evaluating the impact of EEN-specific microbial changes on CD cohorts were essentially study specific. EEN therapy was seen to reduce the abundance of several bacterial genera including *Blautia*, *Subdoligranulum*, *Halomonas*, *Ruminococcus*, *Faecalibacterium*, *Bifidobacteriaceae*, *Lachnospiraceae*, Bacteroidales, and Enterobacterales. Concomitant increases in *Clostridiales*, *Nesterenkonia*, *Rhizobium*, *Gemella*, and *Ruminococcus gnavus* were also reported [64,65,66,67,68,69]. When microbial changes were evaluated based on clinical response vs. non-sustained response, again, specific signatures were study-dependent, but generally, responders had increased levels of Firmicutes including *Faecalibacterium*, *Roseburia*, *Anaerostipes* genera, and *Ruminococcus bromii*, as well as Bacteroidetes including *Parabacteroides* and *Flavonifractor* species. Non-sustained responders had higher abundance of Proteobacteria including *E. coli*, *Blautia*, *Streptococcus*, *Granulicatella*, and *Rothia* genera (Table 3).

When the impact of inulin-type fructan supplementation was assessed, Valcheva et al. showed an increase in Bifidobacteriaceae and Lachnospiraceae abundance along with an increase in colonic butyrate levels in UC patients, but this did not correlate with disease activity scores [51]. GOS prebiotic treatment was associated with a reduction in *Oscillospira* and *Dialister* abundance and an increase in *Alistipes* (Table 3).

## 4. Discussion

The concept of altering the gut microbiota to improve health is now a well-established concept in medicine. Microbiome-based therapies include dietary interventions, prebiotics, probiotics, antibiotics, phage therapy, FMT, live biotherapeutics, and microbiome mimetics. In the context of IBD, understanding the interaction between gut microbes, IBD therapeutics, as well as environmental factors and disease progression is essential if we want to achieve symptom resolution and/or avoid adverse reactions. Several prospective studies have investigated the role of gut microbiota as a potential biomarker for treatment response; yet, a systematic evaluation of microbiota changes following IBD treatment has not been undertaken. Therefore, the aims of this systematic review and meta-analysis were to define the microbial changes following pharmacological and nutritional IBD treatments and identify microbial predictors of treatment response.

The review demonstrated a consistent increase in alpha diversity indices following biological therapies, 5-ASA, and corticosteroids. Additionally, higher microbial diversity at baseline and/or following treatment was shown to be predictive of treatment response in studies on infliximab, adalimumab, and ustekinumab [41,43,57,59]. Increased microbial richness in the gut has been strongly associated with good health, and a less diverse microbiome is consistently implicated in IBD [19,75]. Therefore, a more diverse gut microbiome at baseline might indicate a greater abundance of microbial taxa with anti-inflammatory properties (directly or metabolite mediated). A subsequent reduction in colonic inflammation and preservation of mucosal barrier function/integrity may lead to stronger treatment response. However, the exact mechanisms by which these therapeutics impact microbial diversity remains to be elucidated. The studies included in this review tracked alpha diversity changes for varying periods ranging from 2–30 weeks. Considering that IBD is a life-long disease that is progressive in nature, it would be of significant interest to evaluate if the increase in microbial diversity can be sustained longitudinally and its association with long-term health outcomes. By generating this insight through extended interrogation of large longitudinal IBD cohorts, which comprise a range of disease presentations, as well as ethnic diversity, will contribute to the improvement of IBD treatment results and also IBD management.

Comparison of the studies looking at biologic therapies demonstrated considerable heterogeneity in microbial changes in response to treatment; however, an enrichment of putative SCFA-producing bacteria was consistently noted. SCFAs such as acetate, butyrate, and propionate exert immunomodulatory and anti-inflammatory effects through mechanisms such as the modulation of intestinal wall permeability and reducing oxidative stress and decreasing inflammation through inhibition of mediators such as NF-κB and IL-8 and are primary sources of energy for colonic epithelial cells [76,77,78]. Therefore, the expansion of SCFA-producing bacteria, following certain treatments, highlights that those alterations in microbial metabolic functions could be more important than identifying individual taxonomic changes in the context of IBD. This concept accords with a recent study by Anathakrishnan et al., which observed that there were significantly greater metagenomic alterations in microbial functions than there were changes in microbial composition in both CD and UC cohorts following treatment with the anti-integrin agent, vedolizumab [42]. This highlights the need for future studies to characterise both compositional and functional alterations in the gut microbiota.

Evidence supporting the role of the gut microbiota as a predictor of treatment response is accumulating, with most studies in the systematic review identifying microbial predictors associated with treatment outcomes. An important consideration is understanding that IBD therapeutics can alter microbial profiles, and conversely, the gut microbiota can influence therapeutic outcomes. Interestingly, the observation in three studies on infliximab and vedolizumab that certain baseline signatures are predictive of treatment response or non-response is strongly suggestive that specific gut microbial factors can influence treatment efficacy [39,42,43]. However, the findings were strikingly heterogenous between studies; thus, it was not possible to perform a meta-analysis to evaluate the potential of microbial predictors of treatment response in IBD reported in these studies. Therefore, the microbiota results have to be interpreted with caution, especially since significant variation between studies in terms of subject age, disease subtype and severity, clinical outcome assessments, geography, medications, the length of the study period, and analysis techniques were noted.

The modulation of the gut microbiota by faecal microbiota transplantation (FMT) is now being used to induce remission in UC. Published randomised clinical trials have highlighted that clinical response relies on the enrichment of certain beneficial bacteria including SCFA-producing strains. Interestingly, recent evidence suggests that baseline microbiome profiles are a good predictor of response, with patients achieving a successful outcome having significantly higher microbial richness prior to, during, and after FMT [79]. This further reinforces the potential of baseline or pretreatment microbiome assessment in defining therapeutic outcomes.

There are several limitations in this review. First, the majority of the studies had small subject numbers and, notably, the patients that were included in the detailed longitudinal microbial analysis often represented a significantly smaller subset of the actual cohort. Another significant limitation of the studies included in this systematic review is that most did not account for confounders that are known to impact gut microbiota such as dietary intake and polypharmacy effects. The lack of studies focusing on certain treatments (vedolizumab (n = 1), ustekinumab (n = 1), etanercept (n = 1), 5-ASA (n = 1), antibiotics (n = 2), corticosteroid (n = 2), prebiotic (n = 1), and SCD (n = 2) also limited the comparisons. Additionally, the lack of standardisation in disease phenotyping, microbial analysis methods, defining outcome assessments (clinical/endoscopic response/remission), study periods, and sampling points remain major barriers to understanding the intricate association between the gut microbiota and therapeutics. Overall, the heterogeneity of the gut microbiota findings between different patient populations is challenging to interpret. These limitations must be considered within the design of future studies. This includes the need to focus on well-phenotyped prospective longitudinal patient cohorts, the appreciation of inflammatory and treatment confounders, as well as considering differences due to geography, age, and diet. Finally, harmonisation across studies in terms of establishing a robust standardised scientific methodology is fundamental to validate, reproduce, and ensure the quality of findings in future research.

## 5. Conclusions

In conclusion, this systematic review contributes to the growing appreciation of the relationship between gut microbes and IBD therapeutics; however, defining optimal treatment strategies remains a challenge. The identification of microbial predictors of treatment response is fundamental to achieving precision medicine in IBD; however, there is still a long way to go before these findings can be translated into clinical practice. Further validation in large longitudinal prospective cohorts is needed.

## Figures and Tables

**Figure 1 pathogens-12-00262-f001:**
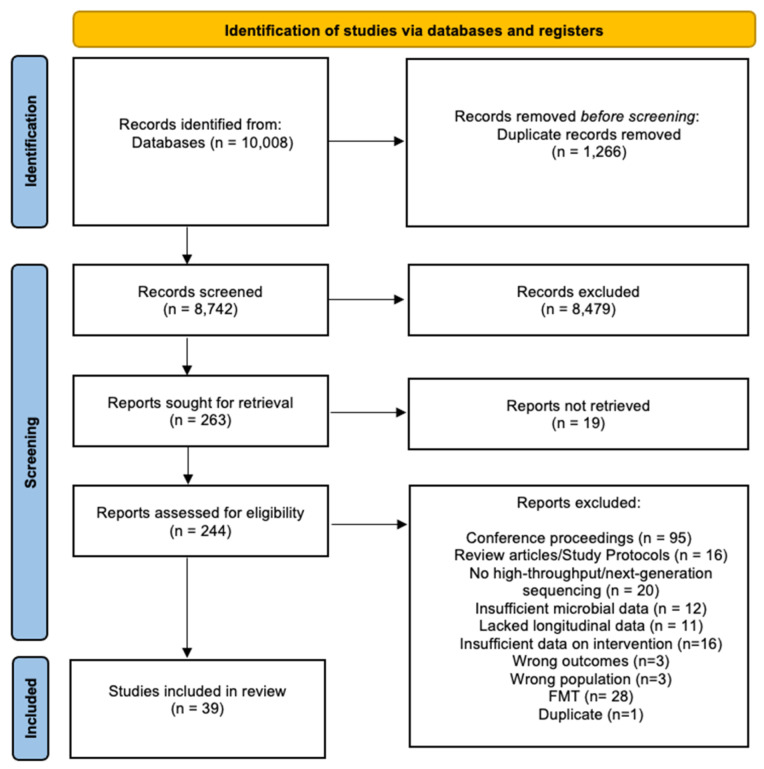
PRISMA diagram showing the flow chart of the included studies.

**Figure 2 pathogens-12-00262-f002:**
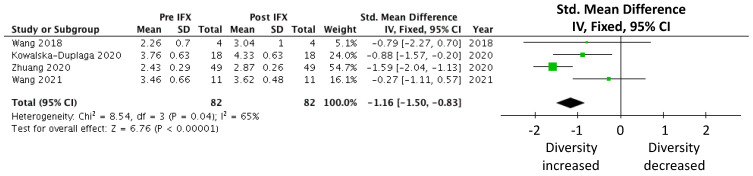
Meta-analysis of alpha diversity (measured by the Shannon index) in CD cohorts following infliximab treatment [38,39,40,54].

**Table 3 pathogens-12-00262-t003:** Microbiota changes following biological therapies, non-biological therapies, and nutritional therapies in studies with CD patients only, UC patients only, or both.

Disease Subtype	Reference	Population	Microbiota Changes due to Treatment
**Biological therapies**			
CD CD CD	Wang et al. [38]	Paediatric (P) CD (n = 4)	Following IFX treatment: Decreased abundance: Enterobacteriaceae, Enterococcaceae, Planococcaceae, and Streptococcaceae. Increased abundance: SCFA-producing genera, *Blautia*, *Faecalibacterium*, *Odoribacter*, and *Sutterella* maintained during therapy. Increased abundance: *Coprococcus*, *Lachnospira*, *Roseburia*, *and Ruminococcus* genera, but their abundances were unstable. Patients with sustained response had higher abundances of SCFA-producing genera, including *Blautia*, *Faecalibacterium*, *Lachnospira*, *and Roseburia* compared to non-sustained responders.
	Zhou et al. [57]	Adult (A) CD (n = 16)	Following IFX treatment: Increased Clostridiales abundance (Lachnospiraceae, Veilonellaceae).
	Dovrolis et al. [59]	A -CD (n = 10)	Following IFX treatment, CD patients: Increased abundance of Proteobacteria and *Rubrobacter* genera. Responders had decreased abundance of Fusobacteria. Responders had increased abundance of Chloroflexi, Ruminococcus_1, *Eubacterium_hallii*, *Eubacterium_eligens*, *Escherichia*, *Shigella*, *and Butyricicoccus* genera. Non-responders had a decreased abundance of *Anaerostipes* genera.
	Kowalska-Duplaga et al. [54]	P -CD (n = 18)	Following IFX treatment: Decreased Actinomycetales abundance. Functional profiling indicated pathways involved in metabolism (protein digestion and absorption, primary and secondary bile acid biosynthesis) and immune response (adipocytokine signalling pathway) were altered following IFX treatment.
	Salamon et al. [55]	P -CD (n = 13)	Following IFX treatment: Decreased abundance of *L. fermentum.*
	Zhuang et al. [40]	A -CD (n = 49)	Following IFX treatment: Increased abundance of Bacteroidetes, Firmicutes (Order Clostridiales, Family Lachnospiraceae): ○ SCFA-producing genera: *Lachnospira*, *Roseburia*, *and Blautia* increased and remained during IFX therapy. Lachnospiraceae and *Blautia* were highly correlated with IFX therapeutic response. Lower abundance of Proteobacteria (Enterobacteriales), Fusobacterium, Enterobacter, and Escherichia-Shigella.
	Wang et al. [39]	P -CD (n = 11)	Following IFX treatment: Increased abundance: *Blautia*, *Clostridium IV*, *Collinsella*, *Eubacterium*, *and Ruminococcus* species. Decreased abundance: *Abitorophia and Lactococcus* species. Patients with sustained response: Increased abundance: *Actinomyces*, *Atopobium*, *and Parabacteroides* genera. Higher abundance: *Methylobacterium*, *Sphingomonas*, *Staphylococcus*, *and Streptococcus* genera at baseline. Patients with non-sustained response: Increased abundance: *Dorea and Holdemania* genera. Higher abundance: *Clostridium* XI and XVIII, *Eggerthella*, *Lachnospiracea incertaesedis*, *Parabacteroides*, *Peptococcus* genera.
	Ribaldone et al. [53]	A -CD (n = 20)	Following ADA treatment: Decreased abundance of Proteobacteria. Proteobacteria decreased in treatment responders. Lachnospiraceae family in patients with normalisation of C-reactive protein levels after ADA therapy.
	Ananthakrishnan et al. [42]	A -CD (n = 42)	Following Vedolizumab treatment: In CD remission patients (n = 10): Decreased relative abundances of *Bifidobacterium longum*, *Eggerthella*, *Ruminococcus gnavus Roseburia inulinivorans*, *and Veillonella parvula* genera between baseline and Week 14 post-treatment. *R. inulinivorans* and Burkholderiales species were more abundant at baseline in patients achieving remission. Significant reduction in 17 pathways at Week 14 compared to baseline, of which 15 (including several tricarboxylic acid pathways (I and V types) and nicotinamide adenine dinucleotide (NAD) salvage pathways) decreased oxidative stress.
	Doherty et al. [41]	A-CD(n = 306)	Following UST treatment, in subjects in remission 6 weeks after induction compared to those with active disease: Bacteroides and *Faecalibacterium* species were significantly more abundant. *Faecalibacterium*, as well as *Blautia*, *Clostridium* XIVa, Ruminococcaceae, and *Roseburia* genera were all present at higher median.
	Sanchis-Artero et al. [61]	CD R (n = 13) CD NR(n = 14)	Following IFX/ADA treatment: In responders, restoration in phyla Bacteroidetes and Firmicutes, in particular those belonging to the class Clostridia, especially the genera *Faecalibacterium*, *Romboutsia*, *Coprococcus*, *Dorea*, *Roseburia*, *Anaerostipes*, *or Lachnospira* genera. In non-responders, there was a significant increase in *Escherichia/Shigella* and decrease in *Faecalibacterium* and *Agathobacter* genera compared to healthy controls. An association was found (*p* < 0.001) in the *F. prausnitzii/E. coli* ratio between responders and non-responders.
CD	Ventin-Holmberg et al. [56]	CD R (n = 12) CD NR(n = 18)	Following IFX treatment: At baseline, Clostridia, Bacilli, *Faecalibacterium*, and *Subdoligranulum* genera were more abundant in responders, whereas Gammaproteobacteria, *Dialister*, and *Anaerostipes* genera were more abundant in non-responders. At 6 weeks, responders had a relative increased abundance of Actinobacteria, Erysipelotrichia, and *Bifidobacterium* and decreased abundance of *Blautia*, *Coprococcus*, *Lachnospiraceae*, and *Dialister* genera. Response was predicted by baseline *Ruminoccocus* species' relative abundance.
UC	**Dovrolis et al. [59]**	A -UC (n = 4)	Following IFX treatment, in UC patients: Responders had increased Bacteroidetes populations and loss of Spirochaetes and Plantomycetes. Responders had significantly increased *Veillonela*, *Tyzzerella*, *Ruminococcus torques*, *Parabacteroides*, *Erisypelatoclostridium. and Bilophila* genera. Non-responders had an increased abundance of *Actinobacteria*, *Porphyromonas*, *Granulicatella.* and *Corynebacterium* genera. LEfSe analysis linked *Bilophila* species to responders and *Granulicatella* species to non-responders.
IBD	Kolho et al. [43]	P -IBD (n = 11)	Following IFX/ADA treatment: Bacteria belonging to Bacilli, Proteobacteria, and low-abundance Clostridium clusters increased in abundance in non-responders. Patients responding to treatment had higher levels of Bifidobacterium, *Clostridium colinum*, *E. rectale*, uncultured Clostridiales, and Vibrio genera and a lower abundance of *Streptococcus mitis* at baseline compared to subsequent non-responders.
	Aden et al. [58]	A-IBD (n = 12)	Following anti-TNF Etanercept treatment: Normalisation of all 14 identified phylotypes (*Coprococcus, R. inulinivorans, Mogibacteriaceae, Erysipelotrichaceae, Ruminococcaceae, Barnesiellaceae, Subdoligranulum variabile, Bilophila, Clostridium, Bacteroides caccae, R. callidus, B. adolescentis, Dorea longicatena*). Increased abundances of *Coprococcus* and R. *inulinivorans* genera during treatment.
	Ding et al. [48]	Adult CD (n = 76) UC (n = 10)	Following anti-TNF-alpha treatment: A decrease in Bacteroidia. No differences between responders and non-responders; most likely due to sample size.
	Olbjorn et al. [50]	P -CD (n = 22) UC (n = 9)	Following anti-TNF-alpha treatment: Decreased abundance: *E. hallii* species.
	Schierova et al. [60]	CD (n = 17) UC (n= 10)	Following IFX/ADA treatment: 13 taxa in CD and 10 taxa in UC were differentially abundant between baseline and endpoints. These differentially abundant taxa were structural zeros, except for the increase of *Ruminococcus* genera in UC.
	Ventin-Holmberg et al. [49]	CD R (n = 13) CD PR (n = 4) CD NR (n = 6) UC R (n = 31) UC PR (n = 8) UC NR (n = 8)	Following IFX treatment: Relative abundance of *Odoribacter*, *Alistipes*, *Butyricimonas*, and *Anaerofilum* genera were lower in non-responders; *Parasutterella*, *Haemophilus*, and *Veillonella* genera were higher in non-responders compared with responders at several time points in all IBD patients. In CD, NRs had lower abundances of SCFA producers from the class Clostridia compared with responders. Odoribacter and unknown Ruminococcaceae genera were significantly increased in Rs compared to NRs; Granulicatella, Enterobacter, and an unknown genus of Peptostreptococcae were increased in NRs compared to Rs. In CD, orders Bifidobacteriales Micrococcales, Lactobacillales, Burkholderiales, and Pseudomonales were significantly more abundant in NRs, whereas Bacteroidales and Desulfovibrionales were significantly elevated in Rs. In UC, the order Bacteroidales, families Enterococcae, Clostridiaceae, Peptostreptococcaceae, and Ruminococcaceae and seven genera differed significantly between groups NR and R.
	Effenberger et al. [62]	CD R (n = 13) CD NR(n = 11) UC R (n = 7) UC NR (n = 5)	Following anti-TNF treatment: Significant decrease of Proteobacteria and associated increase in Bacteroidetes seen in CD responders but not in UC. In CD patients, there were higher Lactobacillus abundances which were associated with non-remission. In CD remission group, the trend of higher Bacteroides abundance at baseline and follow-up time points were noted. In UC cohort, no such associations detected.
IBD	Park et al. [63]	UC (n = 10) CD (n = 9)	Following anti-TNF treatment: Responders had significantly higher baseline levels of Actinobacteria, *Dorea*, *Agathobaculum*, and *Blautia* genera than non-responders. Non-responders had higher baseline levels of Proteobacteria, Enterobacteriaceae, *Odoribacter* and *R. gnavus* genera. Firmicute:Bacteroidetes ratio decreased post-treatment. Increased abundance of Actinobacteria and *Ruminococcus* similar to that of control group. Decreased abundance of Enterococcaceae and *Enterococcus faecium.*
**Non-biological therapies**			
CD	Pigneur et al. [34]	P -CD (n = 4)	Following Corticosteroid treatment: Increased abundance of *Ruminococcus* and decreased *Blautia* genera. Increased Bifidobacterium abundance and SCFA genera *R. intestinalis* and *Eubacterium.*
	Sprockett et al. [36]	P -CD (n = 74)	Following antibiotic treatment: Metronidazole: Increased genera *Enterococcus*, *Streptococcus*, *Klebsiella*, *Bifidobacterium*, *and Escherichia/Shigella*. Reduced genera *B. vulgatus*, *Lachnospiraceae*, *Lachnoclostridium*, *Alistipes*, *F. prausnitzii*, *Blautia faecis*, *A. putredinis*, *B. caccae*, *Terrisporobacter*, *Coprococcus*, *Veillonella*, and *D. formicigenerans*. Metronidazole + Azithromycin: Increased genus *Enterococcus.* Reduced genera *Bifidobacterium*, *Escherichia/Shigella*, *Morganella morganii*, *Parabacteroides distasonis*, *Haemophilus*, *Bacteroides*, *Faecalibacterium*, *Sutterella*, *Erysipelatoclostridium ramosum*, *B. uniformis*, Lachnospiraceae, and Ruminococcaceae.
UC	Hart et al. [35]	P -UC (n = 10)	Following corticosteroid treatment: Increased genera *Blautia*, *Sellimonas*, and uncharacterised Ruminococcaceae. Decreased genera *Granulicatella*, *Haemophilus*, and *Streptococcus.*
	Ishikawa et al. [37]	A -UC (n = 19)	Following amoxicillin, fosfomycin and metronidazole treatment: Increased abundance: Proteobacteria, particularly Enterobacteriaceae. Almost complete depletion: Bacteroidetes.
	Schierova et al. [33]	UC (n = 8)	Following 5-ASA treatment, no changes in faecal microbiome observed.
IBD	Hart et al. [35]	Paediatric CD (n = 4) UC (n = 10)	Following corticosteroid treatment: Increased abundance: *Blautia*, *Sellimonas*, and uncharacterised Ruminococcaceae genera. Decreased abundance: *Granulicatella*, *Haemophilus*, and *Streptococcus* genera.
	Effenberger et al. [62]	CD R (n = 10) CD NR (n = 9) UC R (n = 5) UC NR (n = 5)	Following AZA treatment: Significant decrease of Proteobacteria and associated increase in Bacteroidetes seen in CD responders but not in UC. In CD patients, there were higher Lactobacillus genera abundances, which were associated with non-remission. In CD remission group, trend of higher Bacteroides abundance at baseline and follow-up time points were noted (more pronounced in AZA than anti-TNF group). Klebsiella species was significantly associated with AZA failure in CD patients. In UC cohort, no such associations detected.
**Nutritional therapies**			
CD	Kaakoush et al. [64]	Paediatric CD (n = 5) HC (n = 5)	Following EEN: Reduced number of operational taxonomic units (OTU) in those with remission. Recurrence of CD corresponded with an increase in OTUs. Six families (*Erysipelotrichaceae*, *Ruminococcaceae*, *Lachnospiraceae*, *Streptococcaceae*, *Veillonellaceae*, and *Peptostreptococcaceae*) within the Firmicutes were found to correlate with disease activity in some cases.
	Quince et al. [65]	Paediatric CD (n = 23) HC (n = 21)	Following EEN: Reduction in relative abundance in majority of genera, except Lactococcus. Significant reduction in *Bifidobacterium*, *Ruminococcus*, and *Faecalibacterium*, whose abundance was lower than HC prior to EEN. OTU analysis identified reduction of *Bifidobacterium* and increased in *R. gnavus* in most subjects. Oligotypes of *Lachnospiraceae* decreased in abundance.
	Tang et al. [66]	Paediatric CD remission (n = 17) CD non-remission (n = 8) HC (n = 12)	Following EEN: Increased abundance of 3 genera (*Nesterenkonia*, *Rhizobium*, and *Gemella).* Decreased abundance of genus *Halomonas.* Increased abundance of 11 genera (including *Bacteroides*, *Parabacteroides*, *Flavonifractor*, *Tyzzerella*, *Anaerotruncus*, and *Anaerostipes)* in remission, but not in non-remission group. Genera including *Streptococcus*, *Granulicatella*, and *Rothia)* were decreased in remission group but increased in non-remission group. Compared to HC, genus *Corynebacterium* was more abundant in remission group, but not in non-remission group at baseline. At baseline, *Ruminococcus* genera differed between remission and non-remission group.
	Dunn et al. [67]	Paediatric CD (n = 10) HC (n = 5)	Following EEN: Characteristic changes differed between sustained remission group and non-sustained remission group over 12 weeks: ○ Decreased proportion of Firmicutes in sustained remission group, but increased proportion in non-sustained remission group. ○ Increased proportion of Bacteroidetes in sustained remission group, but decreased proportion in non-sustained remission group. At baseline, non-sustained remission group had a large proportion of Proteobacteria compared to sustained remission group. Sustained remission group had a larger proportion of Firmicutes and smaller proportion of Bacteroidetes.
	Costa-Santos et al. [68]	A -CD (n = 10)	Following EEN: Increased relative abundance of Firmicutes (Clostriadiales order). Decrease relative abundance of Bacteroidetes (Bacteroidales order) and Proteobacteria (Enterobacteriales order).
	Diederen et al. [69]	P -CD (n = 43)	Following EEN: High inter-individual variation in microbiota composition. Significant reduction of *Blautia* and *Subdoligranulum* genera abundance during 6 weeks of EEN, but the reduction was not significant compared to baseline at the end of 6 weeks of EEN. Presence of *Dorea longicatena*, *Blautia obeum*, *Bifidobacterium longum*, and *Escherichia coli* at baseline were associated with NR. After 9 weeks, higher relative abundances of butyrate-producing genera *Roseburia* and *Faecalibacterium* were associated with R. LEfSe analysis associated responders with increased *F. prausnitzii*, *B. adolescentis*, and *R. bromii* and non-responders with increased *E. coli* and four *Blautia* OTUs at 9 weeks.
	Suskind et al. [71]	P -CD (n = 5)	Following SCD: Largely patient specific changes. Increased relative abundance of a *Blautia* species, a *Lachnospiraceae* species, *F. prausnitzii*, *R. hominis*, *R. intestinalis*, *A. hallii*, *and E. eligens* genera in most patients. Decreased relative abundance of *Escherichia coli* and a strain of *F. prausnitzii.*
	Tang et al. [66]	CD (n = 31) HC (n = 12)	Following EEN treatment: 20 genera differed significantly between the 2 CD groups. Genera including *Bacteroides*, *Parabacteroides*, *Flavonifractor*, *Tyzzerella*, *Anaerotruncus*, and *Anaerostipes* were increased in remission. *Nesterenkonia*, *Rhizobium*, and *Gemella* genera were increased in CD. 5 genera including *Streptococcus*, *Granulicatella*, and *Rothia* were decreased in the remission group but increased in non-remission. *Halomonas* was decreased in both groups.
	Jiang et al. [70]	CD (n = 7)	Following EEN treatment: Firmicutes had a higher abundance compared to baseline. *Proteus* levels decreased in abundance compared to baseline. Post-treatment microbiota comprised mostly of Proteobacteria, Firmicutes, Bacteroides, Actinobacteria, and Verrucomicrobia.
UC	Valcheva et al. [51]	A -UC (n = 25)	Following prebiotics (inulin-type fructans): Increased Bifidobacteriaceae and Lachnospiraceae genera abundance, but not associated with improved disease scores. Increased colonic butyrate production.
	Wilson et al. [72]	UC (n = 17)	Following GOS prebiotic treatment: Decreased *Oscillospira* and *Dialister* genera and increased *Anaerostipes* genus. Increased *Bifidobacterium* and *Christenellaceae* genera in those in remission at baseline. Decreased *Dialister* genus in those not in remission.
	Chen et al. [73]	UC (n = 14) HC (n = 14)	Following GMA treatment: *Roseburia* and *Dialister* genera were predominant after 5 sessions. *Fusobacterium* species were predominant after 10 sessions. Firmicutes including *Faecalibacterium* and *Roseburia* increased significantly. *Bacteroides* decreased significantly.
IBD	Olendzki et al. [74]	CD (n = 14) UC (n = 7)	After dietary intervention: Top 10 bacteria with increased abundance in both CD and UC are SCFA-producing bacteria, mostly belonging to the class *Clostridia.* *R. hominis* and *F. prausnitzii* abundance increased the most in CD and UC. *E. eligens* and *B. doreii* were also significantly enriched in CD and UC. *P. distasonis* was significantly decreased in both CD and UC.

Abbreviations: UC, ulcerative colitis; CD, Crohn’s disease; IBD, inflammatory bowel disease; HC, healthy controls; EEN, exclusive enteral; GMA, granulomonocytapheresis; GOS, galactooligosaccharide; R, responders; NR, non-responders; PR, partial responders; SCFA, short-chain fatty acid; SCD, specific carbohydrate diet; 5-ASA, 5-aminosalicylic acid; anti-TNF, anti-tumour necrosis factor; IFX, infliximab; ADA, adalimumab; UST, ustekinumab; LEfSE, linear discriminant analysis effect size.

## Data Availability

Not applicable.

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
