# Peer review of "Assessing the Relationship between the Gut Microbiota and Inflammatory Bowel Disease Therapeutics: A Systematic Review"

_pathogens, 2023, doi:10.3390/pathogens12020262_

Round 1

Reviewer 1 Report (Previous Reviewer 1)

The authors addressed my comment on the alpha diversity, however, my major comment on the meta-analysis is still left out. The section on the meta-analysis is obviously less developed compared to the other sections, with only the alpha diversity measure, but no beta diversity or detailed taxonomy differential abundance analysis. Having "meta-analysis" in the title, I am expecting a formal and thorough representation. Again the template remnant is never removed as stated in the response. 

Author Response

[comment 1] The authors addressed my comment on the alpha diversity,

[our response]: Thank you

[comment 2] however, my major comment on the meta-analysis is still left out. The section on the meta-analysis is obviously less developed compared to the other sections, with only the alpha diversity measure, but no beta diversity or detailed taxonomy differential abundance analysis. Having "meta-analysis" in the title, I am expecting a formal and thorough representation.

[our response] We apologise for any unintended promise of beta diversity or taxonomy meta-analysis. This level of analysis falls very much outside the scope of this project which was a project undertaken by medical students. On reflection, the most sensible step forward will be to remove reference to the meta-analysis within the title.

[comment 3] Again the template remnant is never removed as stated in the response. 

[our response] We apologise as we thought this had been removed, this has been checked.

Reviewer 2 Report (Previous Reviewer 2)

Readers of this article are unable to obtain distinct future prospect of how microbiota analysis will be important in the clinical setting of IBD. Innovative systematic review and meta-analysis should lead certain enhancement of readers insight that can contribute to the improvement of not only IBD treatment results but also the evolution and the development in IBD management.

Author Response

[comment 1] Readers of this article are unable to obtain distinct future prospect of how microbiota analysis will be important in the clinical setting of IBD. Innovative systematic review and meta-analysis should lead certain enhancement of readers insight that can contribute to the improvement of not only IBD treatment results but also the evolution and the development in IBD management.

[our response]: Please note as raised by reviewer 1, that mention of meta-analysis in the title is misleading as we do not delve into beta-diversity or community analysis and have therefore removed reference to meta-analysis in the title. However, we take the point that additional enhancement of readers insight that can contribute to the improvement of not only IBD treatment results but also the evolution and the development in IBD management is now further enhanced within the discussion section.

Reviewer 3 Report (New Reviewer)

Mah et al., did an interesting review in finding the relationship between gut microbiome and IBD therapeutics. The authors have done comprehensive work and this work fits the goal of the journal of Pathogens. This manuscript is well-written. Please see my comments below:

1.      Abstract, There are line numbers inserted throughout the main abstract. Please remove.

2.      Line 82-83, not all species of E. Coli are harmful. Several of them are actually beneficial. Bacteroidetes are harmful outside the gut environment, but beneficial inside gut environment. I would suggest the authors tone down the statement, otherwise, it is incorrect.

3.      Line 255, Table 1: This is a large table with lots of empty spaces. Could the authors improve the presentation of the table? It took 11 pages.

4.      Table 2 and 3, similar issue. This table took 9 pages.

Author Response

We are delighted to submit our manuscript revised according to the reviewer comments below:

  1. Abstract, There are line numbers inserted throughout the main abstract. Please remove.

Our response - this has been done

  1. Line 82-83, not all species of E. Coli are harmful. Several of them are actually beneficial. Bacteroidetes are harmful outside the gut environment, but beneficial inside gut environment. I would suggest the authors tone down the statement, otherwise, it is incorrect.

Our response - this has been done

  1. Line 255, Table 1: This is a large table with lots of empty spaces. Could the authors improve the presentation of the table? It took 11 pages.

Our response - this has been done and Table 1 is reduced to 4 pages

  1. Table 2 and 3, similar issue. This table took 9 pages.

Our response - this has been done, Table 2 is reduced to 7 pages and Table 3 is reduced to 8 pages.

Kind regards

Georgina Hold

Round 2

Reviewer 1 Report (Previous Reviewer 1)

The meta-analysis section is not well developed as a separate section. It would be helpful if evenness and PD diversity were included, as well as the beta diversity and the specific microbiome shifts between the treatment. This could be merged into the 3.3.1 section.

Author Response

We thank the reviewer for their comments and submit the revised manuscript for consideration. Each review point is noted and our specific response to each point included. 

[Point 1] The meta-analysis section is not well developed as a separate section. It would be helpful if evenness and PD diversity were included, as well as the beta diversity and the specific microbiome shifts between the treatment.

[Our response] The meta analysis section was undertaken on alpha diversity values provided within the published manuscripts. Based on what was present, the only scores available for inclusion were Shannon. We obtained these scores from raw data within the manuscript OR through discussion with the various research groups. ON this basis it is not data that is amenable to interrogation with additional alpha diversity indices.

[Point 2] This could be merged into the 3.3.1 section.

[Our response] This has been done.

Kind regards

Georgina Hold on behalf of all authors

Reviewer 2 Report (Previous Reviewer 2)

Readers of this article are unable to obtain distinct future prospect of how microbiota analysis will be important in the clinical setting of IBD. Innovative systematic review and meta-analysis should lead certain enhancement of readers insight that can contribute to the improvement of not only IBD treatment results but also the evolution and the development in IBD management.

Author Response

Thank you for raising this point in your review,

[Point 1] Readers of this article are unable to obtain distinct future prospect of how microbiota analysis will be important in the clinical setting of IBD. Innovative systematic review and meta-analysis should lead certain enhancement of readers insight that can contribute to the improvement of not only IBD treatment results but also the evolution and the development in IBD management.

[our response]:  We feel that additional enhancement of readers insight that contributes to the improvement of not only IBD treatment results but also the evolution and the development in IBD management was further enhanced within the discussion section during the previous revision. If there remains something specific that the reviewer requires, please advise as we feel this has been sufficiently addressed.

Round 3

Reviewer 2 Report (Previous Reviewer 2)

Readers of this article are unable to obtain distinct future prospect of how microbiota analysis will be important in the clinical setting of IBD. Innovative systematic review and meta-analysis should lead certain enhancement of reader’s insight that can contribute to the improvement of not only IBD treatment results but also the evolution and the development in IBD management.

Author Response

As confirmed no requirement to respond to this review was needed.

This manuscript is a resubmission of an earlier submission. The following is a list of the peer review reports and author responses from that submission.

Round 1

Reviewer 1 Report

Identifying microbiome status as markers to predict potential therapeutic responses can provide additional guidance to doctors designing targeted treatments and help IBD researchers to reveal the underlying molecular mechanisms between the gut microbiome and host inflammation. The authors presented a very extensive survey on the perturbance of the microbiome community for IBD patients undergoing biological, non-biological treatment, or nutritional therapy. The inclusion criteria were well-defined and clearly stated. The results covered three major treatments, biological, non-biological, and nutrient therapies, and characterized the gut microbiome changes in alpha diversity and detailed composition changes in various taxonomy levels in the short or long-term responders and non-responders. The SCFA-producing genera increasing with the treatment and beneficial microbiome restoring to the baseline looks consistent with the hypothesis. I agree with the authors that the results should be treated with caution since heterogeneity exists in the studies. My major comment is that the "3.4. meta-analysis" part seems incomplete. The author only included a figure from 4 studies, where the reason for the selection of these studies is unclear. At the end of this section, the authors stated "This section may be divided by subheadings. It should provide a concise and precise description of the experimental results, their interpretation, as well as the experimental conclusions that can be drawn.", however, that ends the section unexpectedly. From the presentation perspective, I felt the authors could use more charts to present the results to highlight the common microbiome shifts and ones that were not agreed upon among the selected studies. Also, the tables could be more uniformly formatted to allow easy filtering on a certain condition or microbiome. For the alpha diversity analysis, the authors can further group the metrics to evenness and richness to elucidate the changes. When stating the changes in specific microbiome composition, the taxonomy level could be mentioned and the effect size or significance should also be included.

Author Response

Identifying microbiome status as markers to predict potential therapeutic responses can provide additional guidance to doctors designing targeted treatments and help IBD researchers to reveal the underlying molecular mechanisms between the gut microbiome and host inflammation. The authors presented a very extensive survey on the perturbance of the microbiome community for IBD patients undergoing biological, non-biological treatment, or nutritional therapy. The inclusion criteria were well-defined and clearly stated. The results covered three major treatments, biological, non-biological, and nutrient therapies, and characterized the gut microbiome changes in alpha diversity and detailed composition changes in various taxonomy levels in the short or long-term responders and non-responders. The SCFA-producing genera increasing with the treatment and beneficial microbiome restoring to the baseline looks consistent with the hypothesis. I agree with the authors that the results should be treated with caution since heterogeneity exists in the studies.

[Our response] – We thank the reviewer for their positive comments and agreement with our caution in interpretation due to study heterogeneity. We consider this an important finding and feel this should be highlighted in order to move the field forward

My major comment is that the "3.4. meta-analysis" part seems incomplete. The author only included a figure from 4 studies, where the reason for the selection of these studies is unclear.

[Our response] – This section has been revised to clarify the reason for selection

We assessed the studies for themes which were amenable for meta-analysis. Due to scarce data, meta-analysis was only carried out on the prospective cohort studies that examined the effects of infliximab on the gut microbiota in CD patients, measured by Shannon index [43-45,60]. Eighty-two subjects were included in the meta-analysis which demonstrated a statistically significant increase in alpha diversity following commencement of infliximab treatment (SMD -1.16 (-1.50, -0.83), p<0.00001; Figure 2). The degree of variance resulting from between-study heterogeneity, reported as I2 was 65%.

At the end of this section, the authors stated "This section may be divided by subheadings. It should provide a concise and precise description of the experimental results, their interpretation, as well as the experimental conclusions that can be drawn.", however, that ends the section unexpectedly.

[Our response] – we apologise, this appears to be a template remnant and has been removed.

From the presentation perspective, I felt the authors could use more charts to present the results to highlight the common microbiome shifts and ones that were not agreed upon among the selected studies.

[Our response] – we thank the reviewer for this comment and had a similar intention during the drafting the manuscript. However, upon attempting to generate such charts realised that it was not a simple task and actually ended up becoming more confusing. This is why we opted for the format we present as we did not want to prioritise any findings over other ones as we felt all findings were equal. We felt that segregating studies based on disease presentation was the most appropriate and therefore would prefer to retain our current format.

Also, the tables could be more uniformly formatted to allow easy filtering on a certain condition or microbiome.

[Our response] – Please see above comment regarding filtering based on microbiome but feel that our colour coding and separation based on disease presentation gives the easiest opportunity to view the study findings

For the alpha diversity analysis, the authors can further group the metrics to evenness and richness to elucidate the changes.

[Our response] – This is a really great suggestion which we have implemented in Table 2. The new manuscript contained a revised table 2 which clearly highlights whether the alpha diversity metrics were evenness/richness or phylogenetic diversity metrics.

When stating the changes in specific microbiome composition, the taxonomy level could be mentioned and the effect size or significance should also be included.

[Our response] – The manuscript has been revised to add this detail.

Reviewer 2 Report

The article entitled “Assessing the relationship between the gut microbiota and inflammatory bowel disease therapeutics: A systematic review and meta-analysis” by Gassandra Mah, et al. demonstrated that distinct microbial signatures are seen following treatment and are associated with treatment response.

This study has some value, however, the mention about the advantage of microbiota analysis in “Discussion” is not enough. As a result, the authors present a limited interest.

Thus, there are areas that need to be improved.

Major comments

1.    The authors should provide the mention about the advantage of microbiota analysis as promising predictive biomarker of treatment response. It is unclear that microbiota analysis is superior to the existing parameters (e.g., C-reactive protein;CRP, erythrocyte sedimentation rate;ESR, endoscopic disease activity index, pathologic findings, etc. ).

2.    Please discuss about ridiculous comparison analysis of pretreatment samples between responders and non-responders to IBD treatment. It is unclear how pretreatment gut microbiota interacts treatment response.

3.    There is not enough mention about available modulation, such as faecal microbiota transplantation(FMT), that can yield reliable improvement of IBD treatment.

Minor comments

1.    In “Discussion”, first part (line 514-517) is needless.

2.    The aim of this study was to conduct a systemic review and meta-analysis to define changes in the gut microbiota following IBD treatment and to identify microbial predictors of treatment response. Therefore, descriptions in the “Introduction” (line 45-87) are somewhat redundant and need shortening considerably.

3.    In “Figures”, authors should not only adjust and reform diagram composition but also change the font size of chart characters because it is too hard to read this manuscript.

Author Response

The article entitled “Assessing the relationship between the gut microbiota and inflammatory bowel disease therapeutics: A systematic review and meta-analysis” by Gassandra Mah, et al. demonstrated that distinct microbial signatures are seen following treatment and are associated with treatment response.

This study has some value, however, the mention about the advantage of microbiota analysis in “Discussion” is not enough. As a result, the authors present a limited interest.

Thus, there are areas that need to be improved.

Major comments

  1. The authors should providethe mention about the advantage of microbiota analysis as promising predictive biomarker of treatment response. It is unclear that microbiota analysis is superior to the existing parameters (e.g., C-reactive protein;CRP, erythrocyte sedimentation rate;ESR, endoscopic disease activity index, pathologic findings, etc. ).

 [Our response] – The reviewers highlight the requirement for systematically reviewing the literature in the microbiome space to allow the field to make an informed decision on the potential of microbiota analysis as a predictive biomarker of treatment response. Whilst we felt this concept was addressed in the introduction, we have developed this further as follows

Similarly, in IBD, the potential role of the gut microbiota as a predictive biomarker for treatment response has been an area of intense research. It is currently unclear whether microbiota analysis is superior to existing parameters including C-reactive protein, erythrocyte sedimentation rate, endoscopic disease activity index, pathologic findings.

  1. Please discuss about ridiculous comparison analysis of pretreatment samples between responders and non-responders to IBD treatment. It is unclear how pretreatment gut microbiota interacts treatment response.

 [Our response] – We respectfully beg to differ. The notion of being able to predict treatment response based on pretreatment microbiome signatures is a very promising avenue of research. We would direct the reviewer to significant advancement in the field of immunotherapy which now utilises pretreatment screening to define therapeutic decision-making. 

  1. There is not enough mention about available modulation, such as faecal microbiota transplantation(FMT), that can yield reliable improvement of IBD treatment.

[Our response] – We have extended this concept within the discussion.

Evidence supporting the role of gut microbiota as predictors of treatment response is accumulating, with most studies in the systematic review identifying microbial predictors associated with treatment outcomes. An important consideration is understanding that IBD therapeutics can alter microbial profiles and conversely, gut microbiota can influence therapeutic outcomes. Interestingly, the observation in three studies on infliximab and vedolizumab that certain baseline signatures are predictive of treatment response or non-response is strongly suggestive that specific gut microbial factors can influence treatment efficacy [43,47,48]. However, the findings were strikingly heterogenous between studies thus, it was not possible to perform a meta-analysis to evaluate the potential of microbial predictors of treatment response in IBD reported in these studies. Therefore, the microbiota results have to be interpreted with caution especially since significant variation between studies in terms of subject age, disease subtype and severity, clinical outcome assessments, geography, medications, length of study period and analysis techniques were noted.

Modulation of the gut microbiota by faecal microbiota transplantation (FMT) is now being used to induce remission in UC. Published randomised clinical trials have highlighted that clinical response relies on enrichment of certain beneficial bacteria including SCFA-producing strains. Interestingly, recent evidence suggests that baseline microbiome profiles are a good predictor of response, with patients achieving successful outcome having significantly higher microbial richness prior to, during and after FMT. This further reinforces the potential of baseline or pretreatment microbiome assessment in defining therapeutic outcomes.

Minor comments

  1. In “Discussion”, first part (line 514-517) is needless.

 [Our response] – This part has been removed

  1. The aim of this study was to conduct a systemic review and meta-analysis to define changes in the gut microbiota following IBD treatment and to identify microbial predictors of treatment response. Therefore, descriptions in the “Introduction” (line 45-87) are somewhat redundant and need shortening considerably.

 [Our response] – This part has been shortened

Inflammatory bowel diseases (IBD), of which the main subtypes are ulcerative colitis (UC) and Crohn’s disease (CD), are chronic immune-mediated inflammatory conditions that require lifelong treatment [1]. A recent global burden of disease study reported an 85.1% increase in globally prevalent IBD cases between 1990 and 2017, with rising prevalence and incidence even in historically low-incidence regions such as Asia [2].

The natural history of IBD is that of quiescent disease interspersed with flares in disease activity.  Collectively, the debilitating life-long physical and psychosocial symptoms impact patient quality of life and are associated with significant economic and healthcare implications [2,3].

Current drug treatment strategies which include 5-aminosalicylates (5-ASA), corticosteroids and immunomodulators, aim to achieve remission and mucosal healing through reducing the inflammatory burden during active disease [4]. However, they do not specifically target the cause of the inflammation. Growing understanding of the complex multifactorial aetiopathogenesis of IBD has led to the development and use of more targeted therapies, specifically biological agents (anti-TNF-alpha antibodies, anti-cytokine monoclonal antibodies and anti-integrin monoclonal antibodies) to treat moderate to severe disease [5].

Faster induction of remission following diagnosis reduces the risk of irreversible intestinal damage. This important concept has been reinforced by recent trials demonstrating that early initiation of biological therapies is associated with favourable outcomes including increased mucosal healing, reduced hospitalisations, surgeries and complications [6-8]. Anti-TNF-alpha agents, including infliximab, are often first-line in the majority of biologic naïve IBD patients [1], however, the efficacy and tolerability of these therapies are highly variable among patients. It is estimated that 33-50% of patients, especially in the elderly population, discontinue anti-TNF therapy, with a further 13-20% of them experiencing yearly loss of response [9-11]. Nutritional therapies including exclusive enteral nutrition (EEN), probiotics, prebiotics, and specific carbohydrate diet (SCD) aim to induce and maintain disease remission but mechanistic insight into the mode of action of nutritional therapy remains in its infancy. Despite the availability of diverse therapeutic options, the lack of head-to-head trials comparing efficacy of different interventions and the failure to predict a patient’s response to a particular treatment makes selecting an effective and timely treatment highly challenging [12]. Consequently, 50% of patients with CD will require surgery and of those, 33% will require multiple surgeries. Similarly, 30% of patients with UC will undergo colectomy [13,14]. Understanding and modifying the course of IBD progression is a prerequisite for improving patient management and outcomes.

IBD is thought to arise from defective immune regulation producing an abnormal response to luminal antigens which enter the intestinal wall via a leaky epithelium thus triggering inflammation in susceptible individuals [15,16]. Genome-wide association studies have established the association of over 235 genetic loci with IBD [17,18]. IBD risk variants often implicate pathways for microbial recognition or handling/clearance strategies. However, the rapid increase in disease burden in the last 100 years refutes the significance of genetic risk and elevates the importance of other environmental factors which have changed more dramatically in line with disease prevalence.

  1. In “Figures”, authors should not only adjust and reform diagram composition but also change the font size of chart characters because it is too hard to read this manuscript.

 [Our response] – We have provided justification for retaining the current meta-analysis composition but have revised the Forest plot to increase the size of the characters